# Genomic characterization of 99 viruses from the bunyavirus families *Nairoviridae, Peribunyaviridae,* and *Phenuiviridae,* including 35 previously unsequenced viruses

**Marylee L. Kapuscinski**[1], **Nicholas A. Bergren**[1], **Brandy J. Russell**[2], **Justin S. Lee**[1], **Erin M. Borland**[1], **Daniel A. Hartman**[1], **David C. King**[1], **Holly R. Hughes**[2], **Kristen L. Burkhalter**[1,2], **Rebekah C. Kading**[1]*, **Mark D. Stenglein**[1]*

1 Department of Microbiology, Immunology, and Pathology, College of Veterinary Medicine and Biomedical Sciences, Colorado State University, Fort Collins, Colorado, United States of America, 2 Arboviral Diseases Branch, Division of Vector Borne Diseases, Centers for Disease Control and Prevention, Fort Collins, Colorado, United States of America

* Rebekah.Kading@colostate.edu (RCK); Mark.Stenglein@colostate.edu (MDS)

**Data Availability Statement:** All sequencing datasets have been deposited into GenBank under

## Abstract

Bunyaviruses (*Negarnaviricota*: *Bunyavirales*) are a large and diverse group of viruses that include important human, veterinary, and plant pathogens. The rapid characterization of known and new emerging pathogens depends on the availability of comprehensive reference sequence databases that can be used to match unknowns, infer evolutionary relationships and pathogenic potential, and make response decisions in an evidence-based manner. In this study, we determined the coding-complete genome sequences of 99 bunyaviruses in the Centers for Disease Control and Prevention's Arbovirus Reference Collection, focusing on orthonairoviruses (family *Nairoviridae*), orthobunyaviruses (*Peribunyaviridae*), and phleboviruses (*Phenuiviridae*) that either completely or partially lacked genome sequences. These viruses had been collected over 66 years from 27 countries from vertebrates and arthropods representing 37 genera. Many of the viruses had been characterized serologically and through experimental infection of animals but were isolated in the pre-sequencing era. We took advantage of our unusually large sample size to systematically evaluate genomic characteristics of these viruses, including reassortment, and co-infection. We corroborated our findings using several independent molecular and virologic approaches, including Sanger sequencing of 197 genome segments, and plaque isolation of viruses from putative co-infected virus stocks. This study contributes to the described genetic diversity of bunyaviruses and will enhance the capacity to characterize emerging human pathogenic bunyaviruses.

## Author summary

Prior knowledge about families of pathogens can enhance efforts to prepare for and respond to emerging disease threats. The CDC's Arbovirus Reference Collection (ARC)

accessions MH484273–MH484350 and MK896421–MK896656 and MK965544. Quality-filtered sequence reads have been deposited in the sequence read archive (SRA) under Bioproject ID PRJNA543521.

**Funding:** This research was funded by the Colorado State University Office of the Vice President for Research (RCK, MDS), the CSU Infectious Disease Research Center (RCK). KB, HRH, and BR are employees of the US Centers for Disease Control and Prevention Division of Vector-Borne Diseases and contributed to study design, data analysis, and manuscript preparation as part of their normal duties. Computational resources were supported by NIH/NCATS Colorado CTSA Grant Number UL1 TR002535 (MDS). The funders had no role in study design, data collection and analysis, decision to publish, or preparation of the manuscript.

**Competing interests:** The authors have declared that no competing interests exist.

comprises a world reference repository of arthropod-borne viruses that are either patho-genic or are related to known pathogens. Many viruses in this collection were isolated before genome sequencing was readily available, resulting in an incomplete understanding of their potential relevance to human or animal health. In this study, we sequenced the genomes of 99 bunyaviruses in the ARC. We performed detailed phylogenetic analyses, described multiple well-supported instances of genome segment reassortment, and identified co-infection in several virus stocks. These sequences enhance the public database of known bunyaviruses which can be interrogated to identify emerging viruses during an outbreak, and contribute to a better understanding of bunyavirus evolution and patho-genic potential.

## Introduction

New pathogens continue to emerge and threaten human, animal, and plant health. For humans alone, an average of two new disease-causing agents are reported each year [1–3]. These pathogens have primarily been viruses, have usually been associated with animal reservoirs, and many are vector-borne [2]. Bunyaviruses (*Riboviria*: *Negarnaviricota*: *Bunyavirales*) rank among the most serious infectious threats to humans (e.g. Crimean Congo hemorrhagic fever virus [4]), animals (e.g. Rift Valley fever virus [5]), and plants (e.g. tomato spotted wilt virus [6]) [7,8]. Bunyaviruses accounted for approximately 20% of the 188 human pathogenic viruses identified between 1901 and 2005 [3], and new pathogenic bunyaviruses continue to emerge [9–13]. Traditionally bunyaviruses have been grouped according to their serological cross-reactivity (i.e. they have been binned into serogroups), but molecular data and the discovery of many new viruses has driven the reorganization of the order. The order *Bunyavirales* now includes 12 families: *Arenaviridae*, *Cruliviridae*, *Fimoviridae*, *Hantavirdae*, *Leishbuviri-dae*, *Mypoviridae*, *Nairoviridae*, *Peribunyaviridae*, *Phasmaviridae*, *Phenuiviridae*,*Tospoviridae*, and *Wupedeviridae* [14,15].

Bunyaviruses have segmented, negative-sense or ambisense RNA genomes. Many currently characterized bunyavirus genomes are composed of three segments: a large (L) segment encoding a protein that functions as an RNA-directed RNA polymerase (RdRp); a medium (M) segment encoding glycoproteins (Gn and Gc); and a small (S) segment encoding a nucleo-protein (NP) and, in some clades, a non-structural protein, NSs. Some bunyavirus lineages have different numbers of genome segments. For instance, the majority of viruses in the *Are-naviridae* family are bisegmented, and some plant-infecting emaraviruses (family *Fimoviridae*) have 8 segments [16].

Like all viruses with segmented genomes, bunyaviruses are capable of reassortment, which appears to have occurred commonly during bunyavirus evolution [17]. Reassortment has happened between relatively distantly related bunyaviruses (intertypic reassortment) and between closely-related co-circulating strains (intratypic reassortment) [17,18], and can have epidemiological significance. For example, Ngari virus is a reassortant bunyavirus associated with outbreaks of hemorrhagic fever [19]. Iquitos and Itaya viruses are emerging reassortant viruses (*Peribunyaviridae*: *Orthobunyavirus*) that have both been associated with human illness in Peru [9,13]; an intra-lineage reassortant strain of Rift Valley fever virus (*Phenuiviridae*: *Phlebo-virus*) was recently isolated from an ill traveler returning home to China [20]; and severe fever with thrombocytopenia syndrome virus (SFTSV) is a newly-recognized, tick-borne phenui-virus that causes severe human disease and for which reassortment has been recognized in strains detected in human patients [21]. Given the emergence potential of bunyaviruses and

the contribution of reassortment to the generation of new virus strains, having the capability to recognize novel and reassortant viruses is paramount to public health preparedness.

The US Center for Disease Control and Prevention's (CDC) Division of Vector-borne Diseases (DVBD), Arboviral Diseases Branch (ADB), maintains an Arbovirus Reference Collection (ARC). The ARC contains viruses that were isolated from arthropod vectors and vertebrate hosts, as well as from human clinical specimens. Isolates have been collected and deposited into the ARC over a 90-year period. As one of the largest collections of arboviruses in the world, the ARC is an invaluable resource for scientists and public health workers. The ARC includes 2,766 unique isolates, of which 835 are bunyaviruses. Some of these viruses have been genetically characterized, but many have not. Genetic characterization of the viruses in the ARC would expand the value and impact of this collection even further as a scientific and public health resource and infuse a significant amount of novel and important molecular data into the public domain for actionable use. Therefore, the objective of this project was to generate additional and novel genome sequence and phylogenetic data for a large group of bunyaviruses lacking genetic characterization to facilitate the rapid identification and characterization of emerging bunyaviruses.

## Methods

### Sample collection

Viruses represented a subset of bunyaviruses catalogued within the ARC [22] (**Tables 1–3**) that spanned most serogroups as well as ungrouped viruses. RNA was extracted from archived tissue culture supernatants or suckling mouse brain preparations using the QIAamp Viral RNA mini kit according to manufacturer's instructions and eluted in 100µl AVE buffer (Qiagen). TURBO DNase was used to remove residual DNA (ThermoFisher), according to the manufacturer's protocol. RNA concentrations were measured fluorometrically using the Qubit RNA HS Assay kit and a Qubit 3 fluorometer (Life Technologies).

### Library preparation and sequencing

The KAPA RNA HyperPrep Kit was used to prepare sequencing libraries from total RNA according to the manufacturer's protocol using half-scale reactions and the Kapa Dual-indexed Adapter Kit (Kapa Biosystems). Pooled libraries were length-selected for 375–500-bp fragments using a BluePippin 2% cassette (Sage Biosciences). If the length-selected concentration was less than 0.5 nM, additional PCR cycles were performed using the KAPA Library Amplification Kit, until the fluorescence reached that of the kit's internal standard #1. Amplified libraries were cleaned using a 1:1.4 ratio of solid phase reversible immobilization (SPRI) beads (Kapa biosystems). Final length-selected and amplified libraries were diluted to 4 nM and the final concentration determined using the KAPA Library Quantification Kit. Paired-end 2x150bp sequencing was performed on an Illumina NextSeq, producing an average of 2.8 million read pairs per dataset.

### Sequence analysis

Reads were demultiplexed and assessed for quality using FastQC [23]. Cutadapt was used to remove low quality and adapter-derived bases [24]. Read pairs with >96% pairwise nucleotide (nt) identity were collapsed using cd-hit-est to remove likely PCR duplicates [25,26]. Host sequences were filtered by mapping reads to the human (NCBI GCRh38) and/or the mouse (UCSC mm10) genomes using the Bowtie2 aligner with parameters,—local—sensitive—score-min C,60,0 using paired reads [27]. Host-filtered reads were assembled into contigs using

**Table 1. Sequenced genomes of 88 viruses belonging to *Peribunyaviridae (Orthobunyavirus* or *Pacuvirus).***

| Sequence Status * | Virus | Abbreviation | Strain (Passage**) | Serogroup | Collection Date | Collection Country*** | Isolation Source**** | Original Virus Description | GenBank Accession |
|---|---|---|---|---|---|---|---|---|---|
| 4 | Abras | ABRV | 75V1183 (P4V1) | Patois | 1974 | Ecuador | *Culex* mosquito | [46] | MK896654 –MK896656 |
| 1 | Acará | ACAV | BeAn 27639 (P2SM1) | Capim | 1961 | Brazil | Mouse | [47,48] | MK896651 –MK896653 |
| 2 | Aino | AINOV | JaNAr 28 (P9SM1) | Simbu | 1964 | Japan | *Culex* mosquito | [49] | MH484276 –MH484278 |
| 4 | Ananindeua | ANUV | BeAn 109303 (P36SM1) | Guamá | 1966 | Brazil | Opossum | [50] | MK896648 –MK896650 |
| 3 | Anopheles B | ANBV | Original | Anopheles B | 1940 | Colombia | *Anopheles* mosquito | [51] | MK896642 –MK896644 |
| 1 | Antequera | ANTV | AG80-226 (V2SM2V1) | Resistencia | 1980 | Argentina | *Culex* mosquito | [52,53] | MK896639 –MK896641 |
| 4 | Babahoyo | BABV | 75V2858 (P2SM2) | Patois | 1975 | Ecuador | *Culex* mosquito | [46] | MH484273 –MH484275 |
| 1 | Bakau | BAKV | MM-2325 (P8SM1) | Bakau | 1956 | Malaysia | *Culex* mosquito | [54] | MK896633 –MK896635 |
| 1 | Bangui | BGIV | DakHB 754 (P9SM1) | Ungrouped | 1970 | Central African Republic | Human | [55,56] | MK896630 –MK896632 |
| 1 | Barranqueras | BQSV | AG80-381 (V1SM3) | Resistencia | 1980 | Argentina | *Culex* mosquito | [52,53] | MK896627 –MK896629 |
| 3 | Batama | BMAV | DakAnB 1292 (P8SM2) | Tete | 1970 | Central African Republic | Bird | [57] | MK896624 –MK896626 |
| 1 | Belem | BLMV | BeAn 141106 (P11SM1) | Ungrouped | 1968 | Brazil | Bird | [58] | MK896621 –MK896623 |
| 1 | Benevides | BVSV = BENV | BeAr 153564 (P?SM2V1) | Capim | 1968 | Brazil | Mouse | | MK896618 –MK896620 |
| 1 | Benfica | BENV = BNFV | 71U344 (P?SM6) | Capim | 1971 | Peru | Hamster | [59] | MK896615 –MK896617 |
| 1 | Bertioga | BERV | 76V25643 (P?SM3) | Guamá | 1976 | Brazil | *Culex* mosquito | [60,61] | MK896612 –MK896614 |
| 4 | Birao | BIRV | DakArB 2198 (P?SM1) | Bunyamwera | 1969 | Central African Republic | *Anopheles* mosquito | [62] | MH484282 –MH484284 |
| 1 | Bobaya | BOBV | DakAnB 2208 (P9SM1) | Simbu | 1971 | Central African Republic | Bird | [56,57] | MW415980-MW415982 |
| 3 | Boracéia | BORV | SPAr 395 (P?SM2) | Anopheles B | 1962 | Brazil | *Anopheles* mosquito | [63] | MK896609—MK896611 |
| 2 | Boracéia | BORV | SPAr 4080 (P?) | Anopheles B | 1965 | Brazil | *Anopheles* mosquito | [63] | MK896606 –MK896608 |
| 4 | Bozo | BOZOV | DakArB 7343 (P?SM2) | Bunyamwera | 1975 | Central African Republic | *Aedes* mosquito | [64] | MH484285 –MH484287 |
| 2 | Bruconha | BRUV | 77V14676 (P?V1SM5) | C | 1976 | Brazil | *Culex* mosquito | [50] | MK896603 –MK896605 |
| 2 | Bunyamwera | BUNV | 46A-122 (V3) | Bunyamwera | 2006 | Kenya | *Aedes* mosquito | [65] | MH484288 –MH484290 |
| 1 | Bushbush | BSBV | TRVL 26668 (P?SM3) | Capim | 1959 | Trinidad & Tobago | *Culex* mosquito | [47,66] | MK896597 –MK896599 |
| 3 | Buttonwillow | BUTV | A 7956 (P?SM3H1SM1) | Simbu | 1961 | USA | Rabbit | [67,68] | MH484291 –MH484293 |
| 4 | Caimito | CAIV | VP 488A (P8V2SM1) | Ungrouped | 1971 | Panama | Sand fly | [69,70] | MK896592 –MK896594 |
| 1 | Cananéia | CNAV | SPAn 64962 (SM4V1) | Guamá | 1976 | Brazil | Mouse | [50] | MK896589 –MK896591 |
| 4 | Caraparú | CARV | BeAn 3994 (P?SM11) | C | 1956 | Brazil | Monkey | [71] | MK896586 –MK896588 |
| 2 | Enseada | ENSV | 78V213 (P?V1SM2) | Ungrouped | 1976 | Brazil | *Culex* mosquito | [50] | MK896583 –MK896585 |
| 4 | Fort Sherman | FSV | 86MSP18 (P2V1) | Bunyamwera | 1985 | Panama | Human | [72] | MH484294 –MH484296 |
| 2 | Gamboa | GAMV | 75V20086 (P?SM1V1) | Gamboa | 1975 | Ecuador | *Aedeomyia squami-pennis* mosquito | [73,74] | MK896574 –MK896576 |
| 2 | Guajará | GJAV | 18315 (P?SM4) | Capim | 1975 | Ecuador | Hamster | [46,60] | MK896571 –MK896573 |
| 1 | Guaratuba | GTBV | 76V25271 (P?SM3) | Guamá | 1976 | Brazil | *Culex* mosquito | [50] | MK896568 –MK896570 |

*(Continued)*

**Table 1.** (Continued)

| Sequence Status * | Virus | Abbreviation | Strain (Passage**) | Serogroup | Collection Date | Collection Country*** | Isolation Source**** | Original Virus Description | GenBank Accession |
|---|---|---|---|---|---|---|---|---|---|
| 3 | Gumbo Limbo | GLV | FE3-71H (P4V1) | C | 1963 | USA | *Culex* mosquito | [75] | MK896565 –MK896567 |
| 4 | Itaquí | ITQV | BeAn 12797 (P? SM4V1SM2) | C | 1959 | Brazil | Mouse | [76,77] | MK896558 –MK896560 |
| 1 | Itimirim | ITIV | SPAn 47817 (SM6) | Guamá | 1976 | Brazil | Rat | [50] | MK896555 –MK896557 |
| 1 | Juan Díaz | JDV | MARU 8563 (P? SM5ppfV2SM1) | Capim | 1962 | Panama | Mouse | [73] | MK896552 –MK896554 |
| 3 | Kaikalur | KAIV | VRC 713423–2 (P7SM1) | Simbu | 1971 | India | *Culex* mosquito | [78] | MH484297 –MH484299 |
| 4 | Kairi | KRIV | TRVL 8900 (P13SM1) | Bunyamwera | 1955 | Trinidad & Tobago | *Aedes* mosquito | [79,80] | MH484300 –MH484302 |
| 1 | Ketapang | KETV | MM 2549 (P5SM2) | Bakau | 1956 | Malaysia | *Culex* mosquito | [54,60] | MK896546 –MK896548 |
| 1 | Las Maloyas | LMV | AG80-24 (P?V1SM4) | Anopheles A | 1980 | Argentina | *Anopheles* mosquito | [52,53] | MK896543 –MK896545 |
| 1 | Lednice | LEDV | 6118 (P3SM2) | Turlock | 1980 | Czechoslovakia | *Culex* mosquito | [81] | MK896540 –MK896542 |
| 2 | Lokern | LOKV | A 10391 (P?SM5) | Bunyamwera | NA | NA | NA | [82] | MH484303 –MH484305 |
| 2 | Lukuni | LUKV | ColAn 57389 (P? SM5) | Anopheles A | 1976 | Colombia | Vertebrate | [66,83] | MK896537 –MK896539 |
| 4 | Madrid | MADV | BT 4075 (P13SM2) | C | 1961 | Panama | Human | [84,85] | MK896531 –MK896533 |
| 2 | Main Drain | MDV | R4680 (SM3V1) | Bunyamwera | 1974 | USA | *Anopheles* mosquito | [86,87] | MH484309 –MH484311 |
| 2 | Main Drain | MDV | 72V2567 (V1SM1V2) | Bunyamwera | 1972 | USA | *Aedes* mosquito | [86,87] | MH484306 –MH484308 |
| 4 | Marituba | MTBV | BeAn 15 (SM3SH1V1SM2) | C | 1954 | Brazil | Monkey | [71,88] | MK896528 –MK896530 |
| 1 | Minatitlán | MNTV | M67U5 (P5SM1) | Minatitlán | 1967 | Mexico | Hamster | [46] | MK896525 –MK896527 |
| 4 | Mirim | MIRV | BeAn 7722 (SM1V1) | Guamá | 1957 | Brazil | Monkey | [89] | MK896522 –MK896524 |
| 1 | Moriche | MORV | TRVL 57896 (P? SM2V1) | Capim | 1964 | Trinidad & Tobago | *Culex* mosquito | [60,90] | MK896519 –MK896521 |
| 2, 3 | M'Poko | MPOV | BA 365 (P?AM1) | Turlock | 1966 | Central African Republic | *Culex* mosquito | [91] | MK896534 –MK896536 |
| 4 | Murutucú | MURV | BeAn 974 (P14SM2) | C | 1955 | Brazil | Monkey | [71] | MK896516 –MK896518 |
| 2 | Nepuyo | NEPV | HB7-451 (P?SM6) | C | 1967 | Honduras | Bat | [92,93] | MK896513 –MK896515 |
| 3 | Nola | NOLAV | DakAr B 2882 (P10SM2) | Bakau | 1970 | Central African Republic | *Culex* mosquito | [58] | MK896510 –MK896512 |
| 4 | Northway | NORV | 234 (P?SM1BHK(1) SM4) | Bunyamwera | 1971 | USA | *Aedes* mosquito | [94,95] | MH484312 –MH484314 |
| 1 | Okola | OKOV | YM 50 (P9SM1) | Tanga | 1964 | Cameroon | *Eretmapodites* mosquito | [58] | MK896507 –MK896509 |
| 4 | Oriboca | ORIV | BeAn 17 (P12V4ppf3V1SM1) | C | 1954 | Brazil | Monkey | [71,88] | MK896501 –MK896503 |
| 1 | Pacora | PCAV | J19 (P31SM1) | Ungrouped | 1958 | Panama | *Culex* mosquito | [58] | MK896498 –MK896500 |
| 1 | Pahayokee | PAHV | FE3-52F (P?BHK(1) SM1) | Patois | 1963 | USA | *Culex* mosquito | [96] | MK896495 –MK896497 |
| 2 | Patois | PATV | BT 4971 (SM7V1) | Patois | 1961 | Panama | Rat | [97,98] | MK896489 –MK896491 |
| 4 | Peaton | PEAV | CSIRO 110 (SM1) | Simbu | 1976 | Australia | *Culicoides* biting midge | [99] | MH484318 –MH484320 |
| 3 | Potosi | POTV | 89–3380 (V3SM2) | Bunyamwera | 2005 | USA | *Aedes* mosquito | [100] | MH484321 –MH484323 |
| 2 | Pueblo Viejo | PVV | E4-816 (V2) | Gamboa | 1974 | Ecuador | *Aedeomyia squami-pennis* mosquito | [74] | MK896484 –MK896486 |
| 1 | Resistencia | RTAV | AG80-504 (V1SM3V4SM3) | Resistencia | 1980 | Argentina | *Culex* mosquito | [52,53] | MK896478 –MK896480 |

*(Continued)*

**Table 1.** (Continued)

| Sequence Status * | Virus | Abbreviation | Strain (Passage**) | Serogroup | Collection Date | Collection Country*** | Isolation Source**** | Original Virus Description | GenBank Accession |
|---|---|---|---|---|---|---|---|---|---|
| 3 | Restan | RESV | TRVL 51144 (SM2SH1SM2) | C | 1963 | Trinidad & Tobago | *Culex* mosquito | [101] | MK896475 –MK896477 |
| 1 | San Juan | SJV | 75V2374 (P?SM3) | Gamboa | 1975 | Ecuador | *Aedeomyia squami-pennis* mosquito | [74] | MK896472 –MK896474 |
| 1 | Santa Rosa | SARV | M2-1493 (SM5) | Bunyamwera | 1972 | Mexico | *Aedes* mosquito | [83] | MH484324 –MH484326 |
| 1 | Santarém | STMV | BeAn 238758 (P6SM1) | Ungrouped | 1973 | Brazil | Rat | [58] | MK896469 –MK896471 |
| 4 | Sedlec | SEDV | Av 172 (P4SM1BHK2) | Simbu | 1984 | Czechoslovakia | Bird | [102] | MH484327— MH484329 |
| 3 | Shokwe | SHOV | SAAr 4042 (P6SM2V1) | Bunyamwera | 1962 | South Africa | *Aedes* mosquito | [62,103,104] | MH484330 –MH484332 |
| 2 | Tacaiuma | TCMV | SpH 32580 (SM3) | Anopheles A | 1975 | Brazil | Human | [84] | MK896460 –MK896462 |
| 1 | Tanga | TANV | MP 1329 (P7SM1) | Tanga | 1962 | Tanzania | *Anopheles* mosquito | [105,106] | MK896457 –MK896459 |
| 2 | Tataguine | TATV | 79V1463 (SM2) | Tanga | 1979 | Gambia | *Anopheles* mosquito | [107] | MK896454 –MK896456 |
| 1 | Telok Forest | TFV | MalP 72–4 (P?SM2) | Bakau | 1972 | Malaysia | Monkey | [58] | MK896451 –MK896453 |
| 2 | Tensaw | TENV | A9-171B (P?SM5) | Bunyamwera | 1960 | USA | *Anopheles* mosquito | [108] | MH484333 –MH484335 |
| 1 | Termeil | TERV | BP 8090 (P3SM2) | Ungrouped | 1972 | Australia | *Aedes* mosquito | [109] | MK896448 –MK896450 |
| 2 | Thimiri | THIV | VRC 66414 (P15SM1) | Simbu | 1963 | India | Bird | [110] | MH484336 –MH484338 |
| 1 | Timboteua | TBTV | BeAn 116382 (SM2V1) | Guamá | 1967 | Brazil | Mouse | [111] | MK896445 –MK896447 |
| 4 | Tinaroo | TINV | CSIRO 153 (P? BHK4SM2) | Simbu | 1978 | Australia | *Culicoides* midge | [112] | MH484339 –MH484341 |
| 4 | Tlacotalpan | TLAV | 61D240 (P9V1) | Bunyamwera | 2005 | Mexico | *Mansonia titillans* mosquito | [113] | MH484342 –MH484344 |
| 2 | Turlock | TURV | USA 847–32 (P4SM2V1) | Turlock | 1954 | USA | *Culex* mosquito | [114,115] | MK896442-MK896444 |
| 2, 3 | Vinces | VINV | 24188 (P?SM4) | C | 1976 | Ecuador | Hamster | [46] | MK896439 –MK896441 |
| 2, 3 | Weldona | WELV | 77V5691 (P4V2) | Tete | 1976 | USA | midge (species unknown) | [116] | MK896433 –MK896435 |
| 1 | Wongal | WONV | MRM 168 (P6SM1) | Koongol | 1960 | Australia | *Culex* mosquito | [117,118] | MK896430 –MK896432 |
| 1 | Wyeomyia | WYOV | Original (P8SM1V1) | Wyeomia | 1940 | Colombia | *Wyeomyia melanocephala* mosquito | [51] | MH484345 –MH484347 |
| 3 | Yaba-7 | Y7V | Yaba 7 (P5SM1) | Simbu | 2005 | Nigeria | *Mansonia africana* mosquito | [119] | MH484348 –MH484350 |
| 4 | Zegla | ZEGV | BT 5012 (P?SM7) | Patois | 1961 | Panama | Rat | [84,98] | MK896421 –MK896423 |

* Sequence status as of October 3, 2019

1 = Completely new genome sequence

2 = New strain sequence information

3 = Addition of missing genome segments or coding complete segments

4 = Sequence information already existed. In almost all of these cases, sequences were deposited by other authors during the course of this study.

** Passage number as defined in ARC metadata

*** Currently-recognized country names

**** Based on ARC metadata compiled by the CDC

**Table 2. Sequenced genomes of 6 viruses belonging to *Phenuiviridae (Phlebovirus and Uukuvirus)*.**

| Sequence Status* | Virus | Abbreviation | Strain (Passage**) | Serogroup | Collection Date | Collection Country*** | Isolation Source**** | Original Virus Description | GenBank Accession |
|---|---|---|---|---|---|---|---|---|---|
| 4 | Bujaru | BUJV | BeAn 47693 (P10SM2) | Sandfly fever | 1962 | Brazil | Rat | [83] | MK896442 – MK896444 |
| 2 | Kaisodi | KASDV | G 14132 (P6SM1) | Sandfly fever | 1957 | India | Ixodid tick | [120,121] | MK896549 – MK896551 |
| 4 | Palma | PLMV | PoTi 4.92 (P?SM3) | Sandfly fever | 1992 | Portugal | Ixodid tick | [122] | MK896492 – MK896494 |
| 2 | Punta Toro | PTV | D-4021A (SM15) | Sandfly fever | 1966 | Panama | Human | [123] | MK896481 – MK896483 |
| 1 | Sunday Canyon | SCAV | RML 52301–11 (SM5V1SM2) | Sandfly fever | 1969 | USA | Argasid tick | [124] | MK896463 – MK896465 |
| 2 | Zaliv Terpeniya | ZTV | LEIV 21C (P7SM2) | Uukuniemi | 1969 | Russia | Ixodid tick | [125] | MK896424 – MK896426 |

* Sequence status as of October 3, 2019

1 = Completely new genome sequence

2 = New strain sequence information

3 = Addition of missing genome segments or coding complete segments

4 = Sequence information already existed. In almost all of these cases, sequences were deposited by other authors during the course of this study.

** Passage number as defined in ARC metadata

*** Currently-recognized country names

**** Based on ARC metadata compiled by the CDC

SPAdes genome assembler [28]. BLASTn was used to taxonomically assign contigs by nucleotide similarity to the highest scoring sequence in the NCBI nt database with an expect (E) value less than 1e-8 [29,30]. Contigs not taxonomically assigned using BLASTn were assigned by protein-level similarity using DIAMOND aligner to query the NCBI protein database with an expect value of 1e-3 [31]. Viral contigs were validated by inspecting alignment of mapped reads to draft assemblies in Geneious v11.0.2 [32].

Draft bunyavirus genome assemblies were manually validated by aligning host-filtered reads to contigs using Bowtie2 as described above and alignments were independently validated by two people. Bases at the ends of genomes with less than 4x coverage of the same base were trimmed. After removing low quality data and duplicate and host reads, an average of 3.1% of read pairs remained. We performed *de novo* assembly to generate draft viral genome segment sequences, which were validated manually by remapping reads to draft assemblies and by Sanger sequencing. Viral reads accounted for an average of 88% of host filtered unique reads, which produced 748-fold mean coverage depth across virus genomes (**S1 Fig**). In all cases, sequences validated by Sanger sequencing matched NGS-generated sequences, confirming assemblies and ruling out sample mix-ups.

All genome sequences have been deposited into GenBank under accessions MH484273–MH484350, MK896421–MK896656, MK965544, and MW415980-MW415982. All sequences deposited in GenBank are coding-complete. Quality-filtered sequence reads have been deposited in the sequence read archive (SRA) under Bioproject ID PRJNA543521.

## Validation of assemblies by sanger sequencing

Independent Sanger sequence confirmation from re-extracted RNA was obtained from at least one genome segment for 71 of 99 viruses (197 segments). This validation was performed to ensure that the sequencing data were derived from the intended virus stocks, especially when

**Table 3. Sequenced genomes of 5 viruses belonging to *Nairoviridae (Orthonairovirus)*.**

| Sequence status* | Virus | Abbreviation | Strain (Passage**) | Serogroup | Collection Date | Collection Country*** | Isolation Source**** | Original Virus Description | GenBank Accession |
|---|---|---|---|---|---|---|---|---|---|
| 4 | Estero Real | ERV | K 329 (P3SM1) | Ungrouped | 1980 | Cuba | Argasid tick | [126] | MK896577 – MK896579 |
| 2 | Hughes | HUGV | Dry Tortugas (P15SM2) | Hughes | 1962 | USA | Argasid tick | [127] | MK896561 – MK896563 |
| 4 | Sapphire II | SAPV | 75V8196 (P4SM1) | Hughes | 1975 | USA | Argasid tick | [128] | MK896466 – MK896468 |
| 1 | Wanowrie | WANV | Ig 700 (P7SM1) | Ungrouped | 1954 | India | Ixodid tick | [58,60] | MK896436 – MK896438 |
| 4 | Yogue | YOGV | DakAnD 5634 (P7SM1) | Yogue | 1968 | Senegal | Bat | [58] | MK896427 – MK896429 |

* Sequence status as of October 3, 2019

1 = Completely new genome sequence

2 = New strain sequence information

3 = Addition of missing genome segments or coding complete segments

4 = Sequence information already existed. In almost all of these cases, sequences were deposited by other authors during the course of this study.

** Passage number as defined in ARC metadata

*** Currently-recognized country names

**** Based on ARC metadata compiled by the CDC

working with such a large sample set. For 54 viruses, sequences from all three genome segments were independently confirmed. Primers for each segment of each virus were designed from multiple sequence alignments of Illumina-generated sequences for viruses in each serogroup, using Geneious v11.0.2 [32]. Previously-published consensus primers were used when available, such as for viruses in the genus *Orthonairovirus* and in the genus *Orthobunyavirus*, phlebotomus fever serogroup [33], Bunyamwera serogroup, and Simbu serogroup [34] (**S1 Table**). Viral RNA was re-extracted from a separate vial of the same lot as the original isolate in a 96-well plate format using a Qiagen Biorobot 9604 (Qiagen, Valencia, CA, USA) according to manufacturer's instructions. Nucleic acids were eluted in 100 μl AVE elution buffer supplied with the extraction kit, and stored at -20˚C. Amplification of viral RNA was performed in both forward and reverse directions using the Qiagen OneStep RT-PCR kit (Qiagen, Valencia, CA, USA) according to manufacturer's instructions and using the appropriate primer pair listed in **S1 Table**. Amplicons were purified using either the column-based QIAquick PCR Purification Kit (Qiagen) for individual samples or in a 96-well plate format using the Mag-Bind Viral DNA/RNA 96 Kit (Omega Bio-tek, Norcross, GA) on a KingFisher Flex System (Thermo-Fisher Scientific, Waltham, MA) according to the manufacturer's instructions. Sanger sequencing services were provided by GeneWiz (South Plainfield, NJ, USA). Sequence files were imported into Geneious v11.0.2 for end-trimming and generation of consensus sequences and aligned to the Illumina-generated sequences in either Geneious v11.0.2 or MultiAlign (Corpet 1988) software.

## Sequence alignments and phylogenetic analyses

To optimally align sequences, different strategies were used specific to the segment and bunyaviral genus. The open reading frame (ORF) encoding the RdRp protein on the L segment was used for all genera. The ORF of the M segment encoding the Gn/Gc glycoproteins was also used for all genera. The NSm ORF was removed from the alignment because outgroups contained a different genomic organization. Additionally, coding regions for the mucin and GP38

domains were removed from the orthonairovirus sequence alignment. S segments were aligned using the ORF encoding the nucleoprotein. Outgroups were used to root and provide polarity characteristics to the phylogenetic trees. Outgroups were chosen so as to not provide excessive gaps and disturbances in the alignments. Orthotospovirus sequences were used as the outgroups for orthobunyavirus and phlebovirus analyses. Orthohantavirus sequences were used as the outgroup for orthonairovirus analyses. All available reference sequences for each genus were pulled from GenBank's RefSeq database and included in phylogenetic analyses. Alignments were performed on amino acid sequences using the Muscle algorithm and manually inspected afterwards using Seaview v4.1 [35,36]. After alignment, sequences were then converted back to their original nucleotide sequences for phylogenetic analysis. Poorly aligned, divergent positions characterized by excessive gaps in the alignment were removed using the Gblocks server under less strict conditions [37]. A coalescent phylogenetic analysis of the L, M, and S segments was conducted for orthonairoviruses, orthobunyaviruses, and phleboviruses: the L segment of pacuviruses was also executed. The analysis was conducted in MrBayes run once for 10 million steps, sampling every 5,000 steps and discarding the first 10% as burn-in [38,39]. The general time-reversible substitution model (GTR+I+$\Gamma_4$) was used after determining this model to be optimal using ModelTest [40]. Convergence was assessed by examining the stationary ln-likelihood and effective sample size (ESS, >200) parameters using Tracer v1.4 (http://tree.bio.ed.ac.uk/software/tracer). All phylogenies were executed on the phylo.org server [41]. Output tree files were analyzed using FigTree v.1.3.1 (http://tree.bio.ed.ac.uk/software/figtree/).

Co-phylogenies were generated using these trees and a co-phylogeny visualization tool, available at https://github.com/stenglein-lab/TreeTangler, and were rooted using the outgroups described above. Prior to generation of co-phylogenies, nodes with support values lower than 0.95 were collapsed to polytomies using TreeGraph2 software [42].

To place the sequences in the context of all available related sequences, we downloaded all full-length L protein sequences annotated under the genera *Orthobunyavirus* (NCBI taxonomic ID [taxid] 11572) and *Phlebovirus* (taxid 11584), and the family *Nairoviridae* (taxid 1980415). We aligned these sequences using the MAFFT aligner, and, because of the large number of sequences, inferred maximum likelihood trees using FastTree software using model parameters -lg -gamma [43,44].

## Co-infection analysis

To search for evidence of possible co-infections, we determined the ratio of the number of reads in each dataset that aligned to a given virus (virus self-mapping reads) to the number of reads that mapped to any bunyavirus protein sequence in the NCBI nr database, including bunyaviruses sequenced in this study. Self-mapping reads were quantified by aligning host-filtered reads to the corresponding assembled genomes using Bowtie2 as described above. Total bunyavirus reads were quantified by aligning host-filtered reads to a database composed of all the protein sequences for bunyavirus-annotated sequences in the NCBI database and from newly assembled genomes in this report, using DIAMOND as above. When the ratio of these two values was less than 0.95, we manually inspected contigs from the dataset to identify sequences of possible co-infecting viruses.

Candidate coinfections were validated using PCR and independent cell cultures. Specifically, PCR primers were designed that would discriminate between putative co-infecting viruses and the primary virus in each isolate. The originally sequenced stock virus from the ARC was also inoculated onto grivet (*Chlorocebus aethiops*) Vero cell (ATCC CCL-81) cultures for virus isolation, and supernatant was collected when cultures exhibited cytopathic effects

(CPE). Approximately $1\times10^6$ cells were pelleted by centrifugation at 23 x g for 10 min at room temperature and then frozen at -80˚C. RNA was extracted from cell pellets by adding 1 ml of Trizol (ThermoFisher) to pellets, pipetting to lyse, and incubating at room temperature for 5 minutes, following the Trizol protocol. cDNA was made using our standard reverse transcription protocol. Specifically, 500 ng of RNA or 5.5 µl of RNA was incubated at 65˚C for 5 minutes and immediately placed on ice. Random 15-mer at 25 µM, 2 µl of 1X Superscript III buffer, 5 mM dithiothreitol, 1 mM each dNTPs (ThermoFisher), and 100 U Superscript III Reverse Transcriptase (Invitrogen) was added to this solution. Reaction mixtures were then incubated at 25˚C for 5 minutes then at 42˚C for 45 minutes, and finally at 70˚C for 15 minutes with a 4˚C hold. The resulting cDNA was then diluted 1:10 in water. Using segment-specific primers (**S1 Table**), qPCR was performed using LUNA Universal qPCR Master Mix (NEB M3003L) according to the manufacturer's protocol.

In an attempt to separate and isolate the individual co-infecting viruses (see results), the original Abras isolate was plaqued on Vero cells. Ninety individual plaques were picked for RNA extraction and PCR-confirmation using segment-specific primers (**S1 Table**). Single virus stocks were cultured from plaques that were Sanger sequence-confirmed to have only one of the infecting genotypes present.

## Results

### Samples and sequencing

Coding-complete genomes were generated for 99 bunyaviruses [45]. We focused on viruses classified in the *Peribunyaviridae* (orthobunyaviruses), *Nairoviridae* (orthonairoviruses), and *Phenuiviridae* (banyangviruses and phleboviruses). These viruses constituted a globally and phylogenetically diverse set. The isolates were originally collected from vertebrate and arthropod hosts from 27 countries on six continents over the span of seven decades, from 1940 to 2005; **Fig 1** and **Tables 1–3**). Most samples were collected during the 1950s, 60s, and 70s, and 67% of samples originated from North and South America (**Fig 1B and 1C**). Isolates originated from a variety of invertebrate and vertebrate hosts, though *Aedes*, *Anopheles*, and *Culex* mosquitoes were the predominant source of samples (**Fig 1D**). We focused on viruses for which there was limited existing sequence information at the initiation of this project (**Fig 1E** and **Tables 1–3**). Viruses had diverse passage histories. Some viruses were passaged through grivet (*Chlorocebus aethiops*) Vero or hamster (*Mesocricetus auratus*) BHK-21 cell cultures, others were passaged in suckling laboratory mice. Passage histories have been noted in **Tables 1–3**. More information on these viruses can be accessed through the Arbovirus Catalog (https://www.cdc.gov/arbocat) or the ARC website (https://www.cdc.gov/ncezid/dvbd/specimensub/arc/index.html), however taxonomy may vary between these sources and ICTV as taxonomic updates are instituted.

At the time of submission, 203 of the genome segment sequences we generated shared less than 97% pairwise nucleotide identity with existing sequences in the NCBI nucleotide database, as measured by blastn (**Fig 1E**). For 35 of the viruses, all 3 segments shared <85% nucleotide identity with existing sequences by blastn alignment. These 35 viruses represent previously unsequenced viruses (category 1 viruses in **Tables 1–3 and Fig 1E**). For 37 of the viruses, different strains of the same viruses had been previously sequenced or existing sequences were not complete (e.g., only one segment had been sequenced; categories 2 and 3 in **Tables 1–3**). For 26 of the viruses, coding complete genomes already existed (category 4 in **Tables 1–3**). In almost all of these 26 cases, sequences had been deposited by other groups during the course of our study.

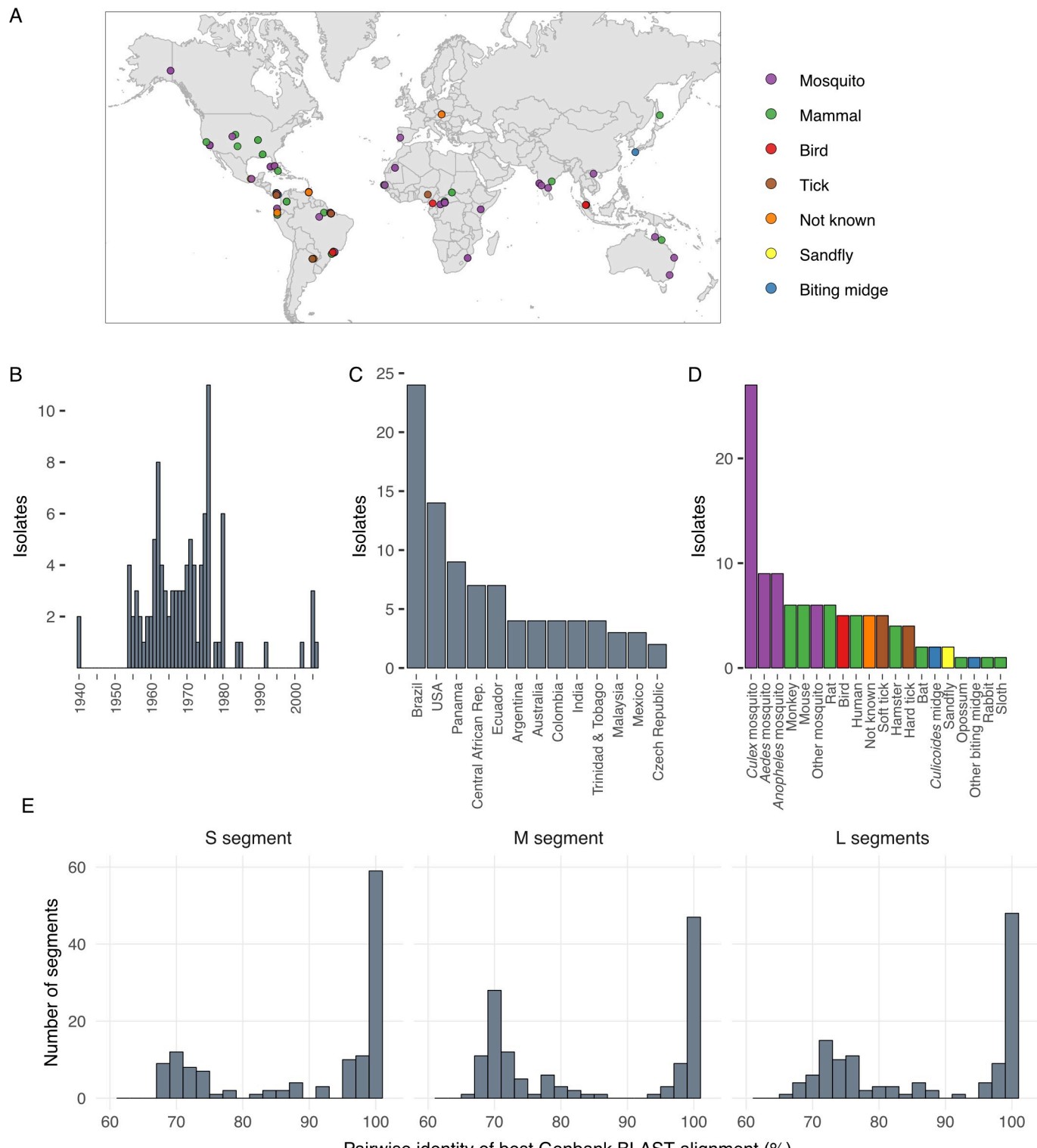

**Fig 1. Characteristics of sequenced bunyavirus isolates.** Bunyavirus isolates were collected from around the world over the span of 7 decades from a variety of vertebrate and arthropod hosts. (A) Map showing original collection location of isolates, color-coded by host type, NK (not known); (B) histogram showing the number of isolates collected each year; (C) histogram showing the number of isolates from the indicated countries, for countries with 2 or more isolates; (D) histogram showing the number of isolates from the indicated host type. (E) histograms showing the pairwise percent nucleotide identity for the highest scoring BLASTn alignments to existing sequences in the NCBI nucleotide database for the sequenced S, M, and L segments.

Several viruses with nearly identical sequences had been given different names and, conversely, some viruses with the same name had quite different sequences. For instance, wongal virus had previously been classified with koongol virus in the species *Koongol orthobunyavirus*, but no sequence information for wongal virus was available [14]. Both viruses were isolated from *Culex annulirostris* mosquitoes several days apart from the same location in Australia [117]. We found that these viruses shared >99.7% pairwise nucleotide identity across all three segments. Thus, koongol and wongal viruses are the same virus. Similarly, all three segments of the Santa Rosa and Lokern (strain A 10391) viruses that we sequenced shared >98.5% identity.

In contrast, the Lukuni virus isolate that we sequenced (strain ColAn 57389) was unexpectedly different from the previously sequenced Lukuni virus (strain TRVL 10076)[129]. These two viruses shared <70% pairwise nucleotide identity across all three segments. Instead, the L and S segments of Lukuni virus strain ColAn 57389 were more similar to those of the Las Maloyas virus isolate that we sequenced (85% and 87% pairwise nucleotide identity). Likewise, the Guajará virus that we sequenced (strain 18315) and the previously sequenced Guajará virus (strain BeAn 10615) shared less than 78% pairwise nucleotide identity over all three segments [129]. These unexpected differences highlight the utility of sequencing additional strains of already-sequenced viruses.

## Phylogenetic analyses

We performed a coalescent phylogenetic analysis of the L, M, and S segments (**Figs 2–6**). Trees included all of the viruses that we sequenced and all of the bunyavirus sequences from the genera in question in the NCBI RefSeq database as phylogenetic landmarks. We also created trees containing all available sequences annotated under the relevant taxa in the NCBI taxonomy database **S2–S4 Figs**). New orthobunyavirus genome sequences fell throughout the *Orthobunyavirus* genus (**Figs 2–4 and S2–S4**). The numbers of genome sequences sequenced in this study from the orthobunyavirus serogroups included: Anopheles A (3), Anopheles B (3), Bakau (4), Bunyamwera (16), Capim (7), Wyeomyia (1), Turlock (3), Koongol (1), Gamboa (3), Guama (7), Minatitlán (1), Group C (11), Patois (5), Simbu (9). Additionally, 5 viruses sequenced fell within the family *Nairoviridae*, 7 in the family *Phenuiviridae* and 2 in the genus *Pacuvirus* (**Figs 5 and 6 and S5**).

Several virus genomes that were sequenced as part of this study had not been assigned to serogroups. We compared the phylogenetic placement of these viruses to that of viruses that had been categorized into established serogroups (**Tables 1–3**). Bangui virus was placed in L and S segment trees within an undesignated clade just prior to the divergence of the Anopheles A and B serogroup (**Figs 2–4**). The Belem, Pacora, and Brazoran viruses formed a monophyletic clade on all three orthobunyavirus trees (**Figs 2–4**), indicating a common ancestor between the three viruses. However, this clade was placed in different locations on the phylogenies for the different segments. On L and M segment trees, these three viruses branched basally to the Patois, Guama, Minatitlán, Group C, and Capim serogroups with strong posterior probability support (**Figs 2 and 3**). The S segment phylogeny showed this clade diverging from all other orthobunyaviruses with posterior probability support of 1.0 (**Fig 4**).

Enseada virus exhibited a similar discrepant placement in the 3 orthobunyavirus trees. The L segment of Enseada virus diverged off of the Patios serogroup (**Fig 2**). The M segment of Enseada virus diverged prior to the bifurcation of the Guama, Capim, Patois, and Minatitlan serogroups (**Fig 3**). Finally, the S segment of Enseada diverged just prior to the Group C serogroup (**Fig 4**). All of these placements were well-supported, with posterior support of 1.

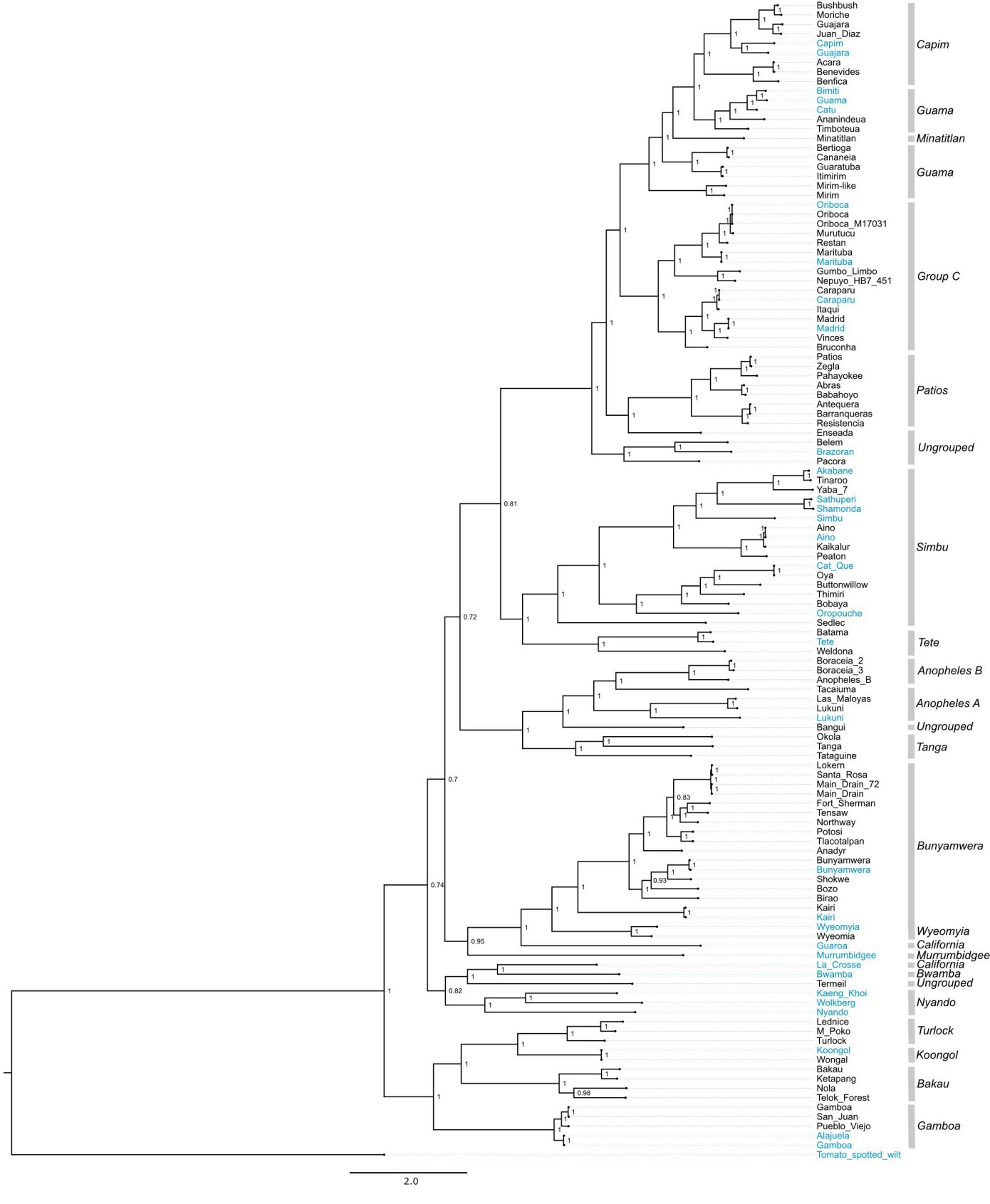

**Fig 2. L segment phylogeny for viruses in *Orthobunyavirus*.** Bayesian phylogenetic tree using the ORF on the L segment. Numbers at nodes indicate posterior probability values. Reference sequences (sequences present in the NCBI RefSeq database) are included as phylogenetic landmarks and colored blue. Sequences from this study are colored black. Classical serogroup distinctions are indicated. Scale bars represent substitutions per site.

Across all segments, Termeil virus branched basal to the California encephalitis and Bwamba serogroups. On the S segment phylogeny, Termeil virus grouped with Murrumbidgee virus (aka Trubanaman virus [130,131]), though posterior probability support was low (**Fig 4**).

The placement of two viruses in our trees diverged from their expected placements based on assigned serogroup classification. Tacaiuma virus has been assigned to the Anopheles A group. However, in the L, M, and S segment phylogenies, Tacaiuma virus instead clustered with high support with viruses in the Anopheles B serogroup (**Figs 2–4**). Guaroa virus has been assigned to the California serogroup, but did not cluster with La Crosse virus as this would suggest. Instead, across all segments, Guaroa virus branched basally to the Wyeomyia serogroup (**Figs 2–4**).

Caimito virus and Santarem virus grouped together with viruses in the genus *Pacuvirus* (**S5 Fig**). Our results coincide with the recent proposal to reclassify Caimito virus as belonging to the genus *Pacuvirus* (*Peribunyaviridae*) [132]. Our phylogenies consistently placed Santarem virus with Caimito virus. We propose Santarem be classified in the *Pacuvirus* genus along with Caimito virus.

Among the nairoviruses, Wanowrie consistently fell within the Qalyub serogroup of the *Orthonairovirus* genus, most closely related to Tăchéng tick virus, though there was insufficient support in the S segment tree to group these two viruses with the rest of those in the Qalyub serogroup (**Fig 5**). Estero Real virus was consistently placed within the Hughes serogroup for all three segments with strong posterior probability support (**Fig 5**).

In phenuivirus trees, Bujaru and Punta Toro viruses consistently grouped within the Sandfly fever serogroup, as expected (**Fig 6**). Also, expectedly, Zaliv Terpeniya and Sunday Canyon grouped within the Uukuniemi serogroup and Palma grouped with Bhanja serogroup across all segments (**Fig 6**). One deviation of note within the phenuivirus trees is the divergent placement of Arumowot virus S segment (**Fig 6**).

## Reassortment

To further evaluate reassortment among the viruses we sequenced, we created co-phylogenies (tanglegrams) of L, M, and S segments (**Figs 7–11**). Prior to generation of these co-phylogenies, we collapsed interior nodes with support values < 0.95 so that poorly supported branching patterns would not produce false signals of reassortment. We also performed all possible pairwise global alignments of the coding regions of the segments to identify pairs of viruses with differential nucleotide identities between segments, for instance, pairs of viruses with closely related L and S segments but relatively divergent M segments (**S2 Table**). Numerous local topological incongruencies were apparent in the trees, but here we focus on the clearest examples of reassortment that were well supported by both phylogenetic discordance and large discrepancies in pairwise genetic distances.

These analyses identified evidence of widespread reassortment among the orthobunyaviruses (**Figs 7–9**). In some cases, these corresponded to reassortment events that had been previously identified, but in other cases these were newly described. Most, but not all, well supported instances of reassortment involved viruses with similar L and S segments but relatively different M segments. This was evident in the relatively similar topologies of the L and S segment trees and the increased degree of crossing-over in the L-M and M-S tanglegrams. This corresponds to reassortant progeny that comprise the L and S segment of one co-infecting

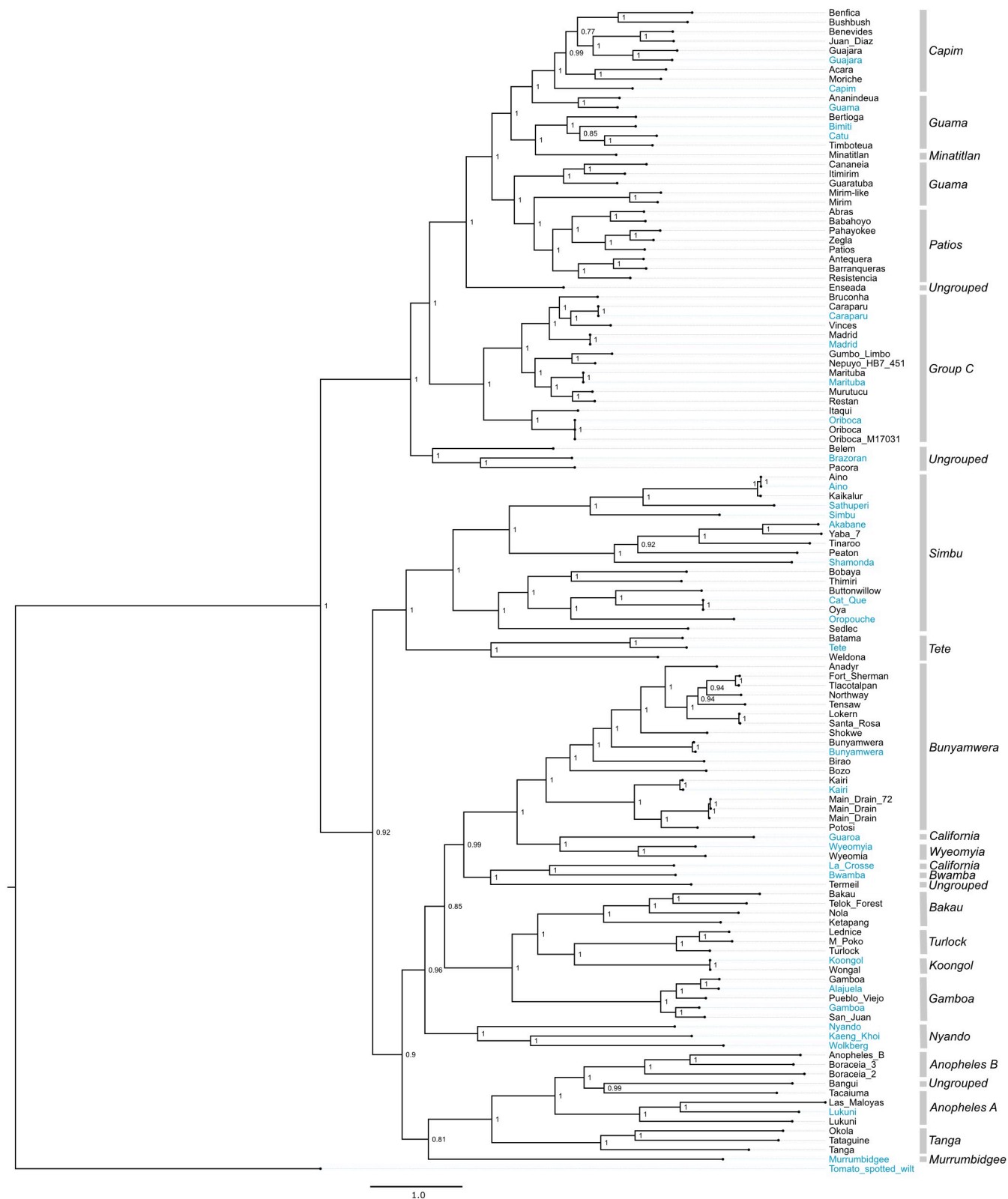

**Fig 3. M segment phylogeny for viruses in *Orthobunyavirus*.** Bayesian phylogenetic tree using the ORF encoding Gn/Gc on the M segment. Numbers at nodes indicate posterior probability values. Reference sequences are colored blue and sequences generated in this project are colored black. Classical serogroup distinctions are indicated. Scale bars represent substitutions per site.

parent virus and the M segment of the second parent. In contrast, the differential placement of Brazoran, Belem and Pacora viruses on the L and M as opposed to the S segment phylogenies supports the hypothesis that this group of viruses resulted from an ancestral reassortment event involving the replacement of an S segment [12].

Consistent with previous reports, there was evidence of extensive reassortment involving all 3 segments in the Bunyamwera serogroup [19,133,134]. The Fort Sherman and Anadyr virus isolates sequenced in this study have S segments with 90% pairwise nucleotide identity but relatively different L and M segments that shared <75% pairwise nucleotide identity. This is one of the few examples we identified of viruses with closely related S segments but relatively different L and M segments. The S and L segments of Potosi virus were most closely related to those of Tlacotalpan virus, but the M segment was more related to Kairi and Main Drain viruses (**Figs 2–4 and 7–9 and S2 Table**). Both Main Drain isolates that we sequenced had S and L segments that were >95% identical to those of Santa Rosa and Lokern viruses. But the Main Drain M segments clustered instead with those of Potosi and Kairi viruses. Shokwe and Bunyamwera viruses L and S segments grouped together, but their M segments did not form a monophyletic cluster.

In the Bakau serogroup, Ketapang and Bakau viruses had similar L and S segments (81 and 88% identical) but relatively different M segments (66% identical). The M segment of Bakau virus was instead more closely related to that of Telok Forest virus, with which it shared 70% pairwise nucleotide identity.

The two Boracéia virus strains that we sequenced (SPAr 395 and SPAr 4080) were isolated from *Anopheles cruzii* mosquitoes in São Paulo, Brazil in 1962 and 1965, respectively (**Table 1**). These isolates had closely related L and S segments with >91% pairwise nucleotide identity but M segments that shared only 63% identity (**Fig 7 and S2 Table**). The Bangui virus L and S segments branched basally to the viruses in the Anopheles A and B groups in the L and S segment trees, but in the M segment tree, Bangu virus fell on a well-supported branch with Tacaiuma virus nested within these groups. Patois and Zegla viruses had nearly identical L and S segments with 97% and 99.7% pairwise nucleotide identity, but their M segments shared only 73% identity.

In the Simbu group viruses, Aino, Kaikalur and Peaton viruses formed a well-supported cluster on L and S segment trees, but the M segment of Peaton virus was at the end of a long, separately placed branch. The Tinaroo and Akabane virus L and S segments were 87 and 95% identical, but the M segments shared only 65% pairwise nucleotide identity. A similar pattern was evident for Athuperi and Shamonda viruses.

Antequera, Barrenqueras, and Resistencia viruses exhibited patterns of sequence relatedness indicative of multiple past reassortment events. The S segments of these three viruses were all ≈99% identical. The L segments of Antequera and Barrenqueras were also nearly identical (99%), but these were both only ≈90% identical to the L segment of Resistencia virus. The M segments of the three viruses all shared <71% pairwise nucleotide identity.

In the Guama and Capim group viruses, there were several examples of apparent M segment reassortment. Bertioga and Cananeia viruses had highly related L and S segments but M segments that were only 67% identical. The same pattern was evident for Itimirim and Guaratuba viruses, for Moriche and Bushbush viruses, for Acara and Benevides viruses, for Guajara and Juan Diaz viruses, and for the trio of Bimiti, Guama and Catu viruses.

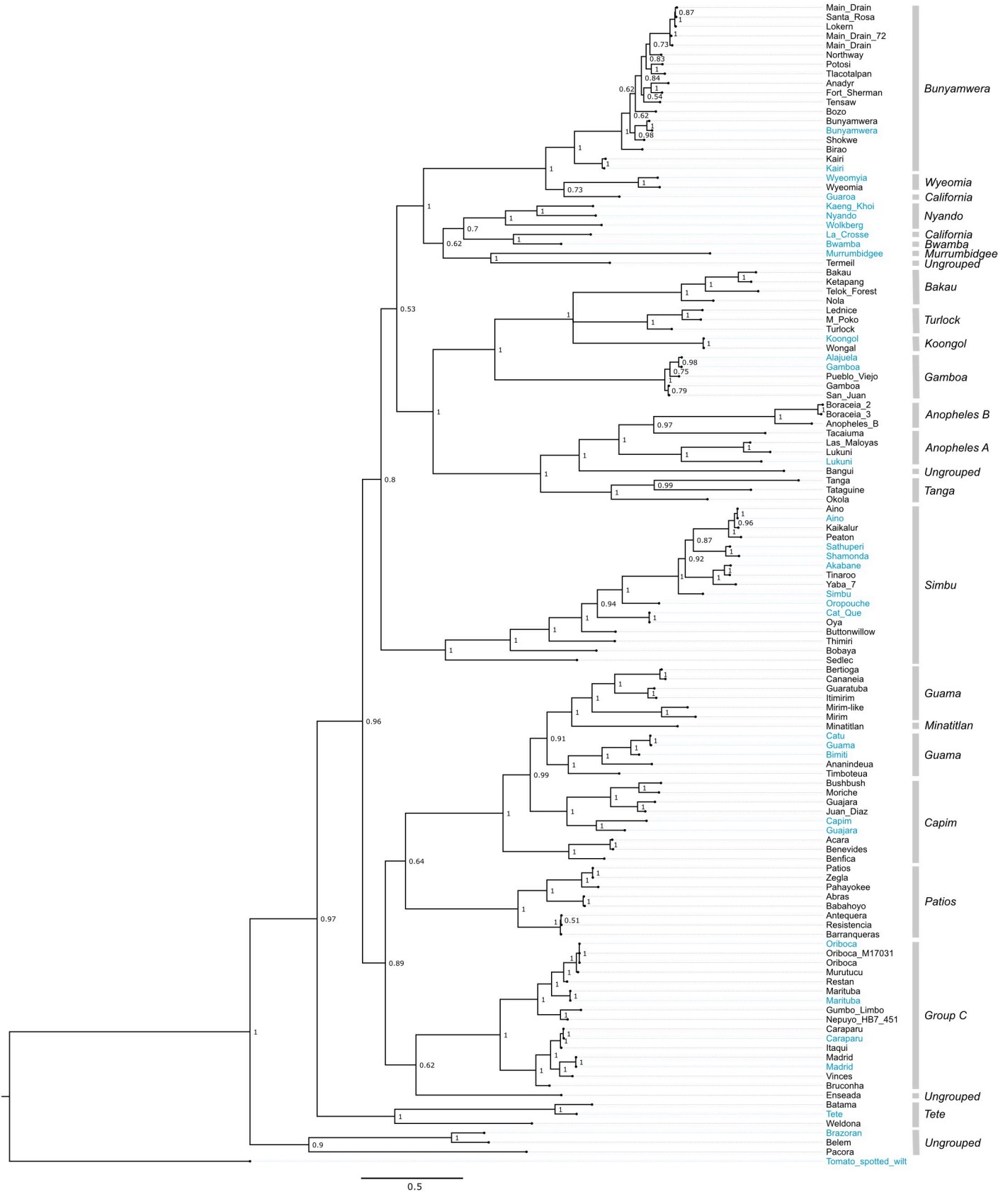

**Fig 4. S segment phylogeny for viruses in *Orthobunyavirus*.** Bayesian phylogenetic tree using the ORF encoding NP on the S segment. Numbers at nodes indicate posterior probability values. Reference sequences are colored blue and sequences generated as part of this study are colored black. Classical serogroup distinctions are indicated. Scale bars represent substitutions per site.

Reassortment has also clearly shaped the evolution of the Group C orthobunyaviruses, as has been previously noted [17,135,136]. Caraparu and Itaqui viruses had nearly identical L and S segments but quite different M segments. The M segment of Itaqui virus was instead much more similar to that of Oriboca virus. The L and S segments of Vinces virus clustered with those of Madrid virus, but the Vinces virus M segment was more closely related to that of Caraparu virus. Restan, Murutucu, and Oriboca virus formed a well-supported cluster on L and S segment trees, but on the M segment tree, Restan amd Murutucu viruses remained clustered together while the Oriboca virus M segment was on a branch basal to the other Group C viruses with Itaqui virus.

There was little evidence for reassortment among the orthonairoviruses in our analysis. The only instance of possible reassortment was for Hughes virus. Despite similar pairwise nucleotide identities between the three segments of Hughes, Sapphire II, and Estero Real viruses, these three viruses had different branching patterns on the three segment trees, albeit with

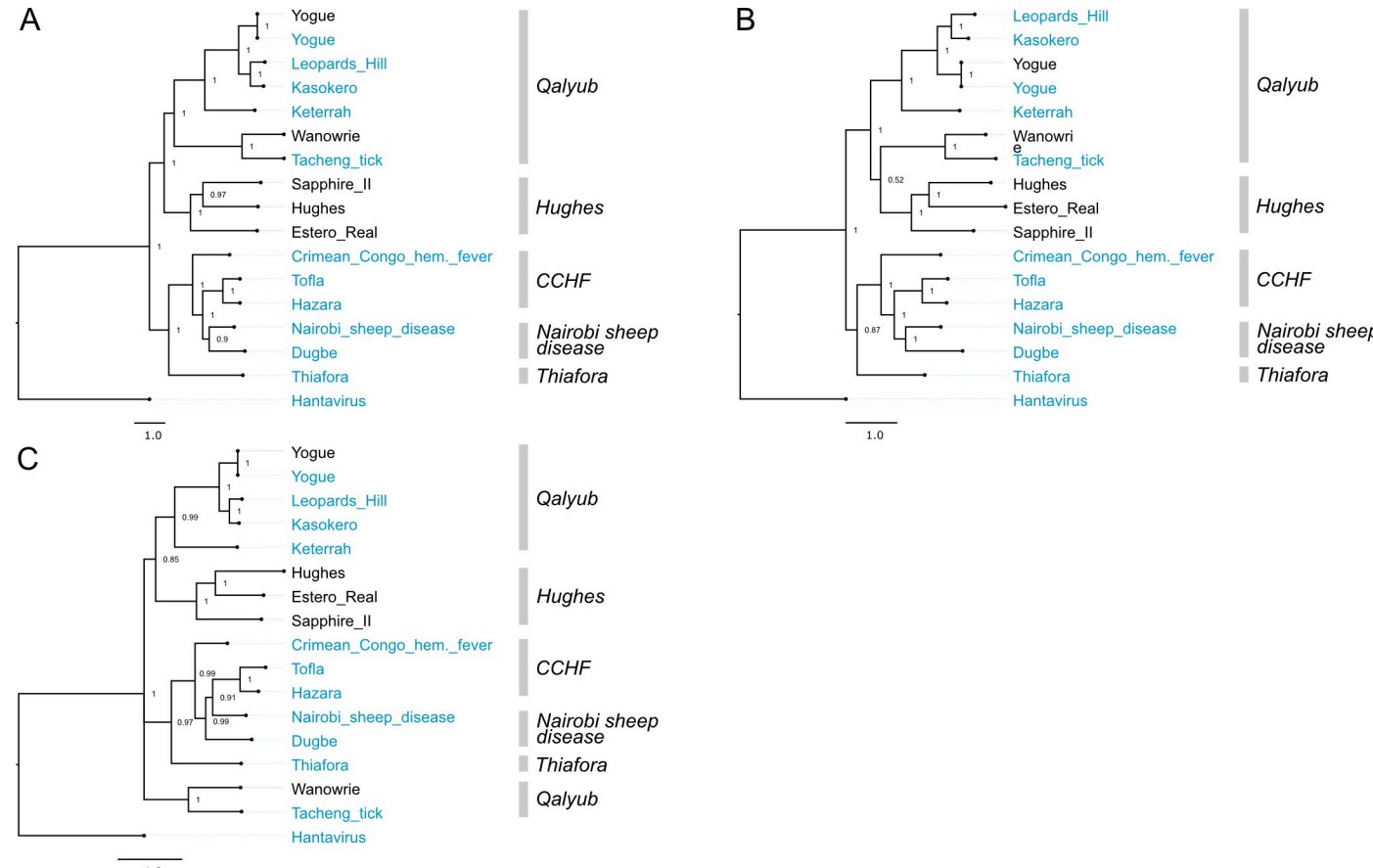

**Fig 5. Phylogenetic analysis of viruses in *Nairoviridae*.** Bayesian phylogenetic tree using (A) the ORF encoding the RdRp on the L segment, (B) the ORF encoding Gn/NSm/Gc on the M segment, and (C) the ORF encoding NP on the S segment. Numbers at nodes indicate posterior probability values. Reference sequences are included as phylogenetic landmarks and colored blue. Sequences generated in this study are colored black. Classical serogroup distinctions are indicated. Scale bars represent substitutions per site.

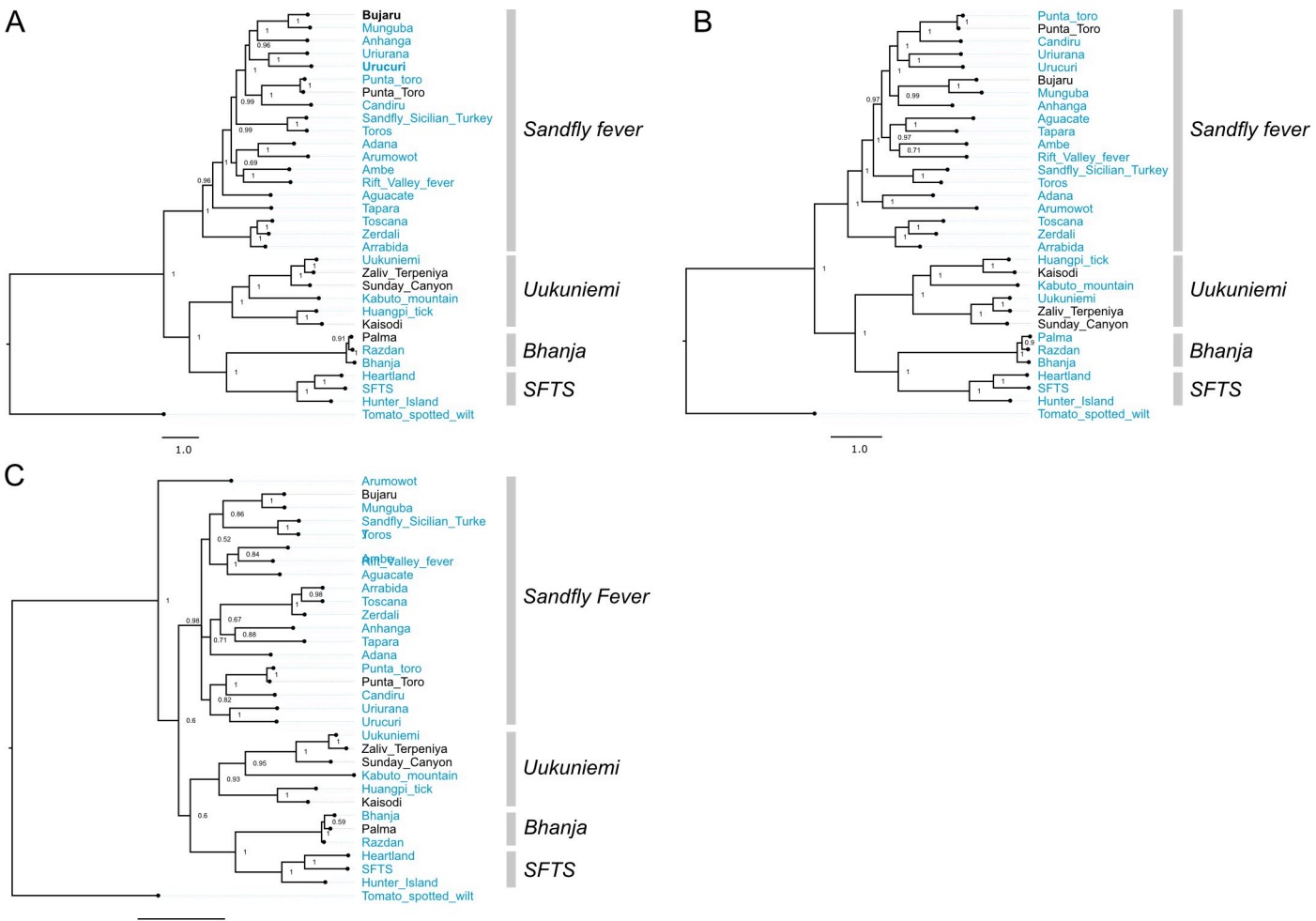

**Fig 6. Phylogenetic analysis of viruses in *Phenuiviridae*.** Bayesian phylogenetic tree using (A) the ORF encoding the RdRp on the L segment, (B) the ORF encoding Gn/Gc on the M segment, and (C) the ORF encoding NP on the S segment. Numbers at nodes indicate posterior probability values. Reference sequences are included as phylogenetic landmarks and colored blue. Sequences generated in this study are colored black Classical serogroup distinctions are indicated. Scale bars represent substitutions per site.

long branch lengths and support values <1 (**Fig 10 and S2 Table**). In phlebovirus tanglegrams, Aguacate and Tapara viruses showed evidence of reassortment, but there was no strong evidence for reassortment among the phleboviruses sequenced in this study (**Fig 11**).

## Co-infection

We identified evidence of co-infection (i.e. RNA from more than one virus) in four virus stocks: Abras virus, Guaratuba virus, Hughes virus, and one of the two Main Drain virus isolates. The initial indication that there might be co-infection in these stocks was the assembly of more than the expected three bunyavirus contigs from the corresponding datasets, and was followed up with additional computational and molecular tests to validate possible co-infections.

In the Abras virus dataset, six coding-complete sequences assembled, including 2 L, 2 M, and 2 S segments. Three of the segments matched the previously sequenced Abras virus isolate 75V1183 (the same isolate that we sequenced) with 99–100% nucleotide identity [137]. The other three segments belonged to a previously unsequenced bunyavirus in the Guama

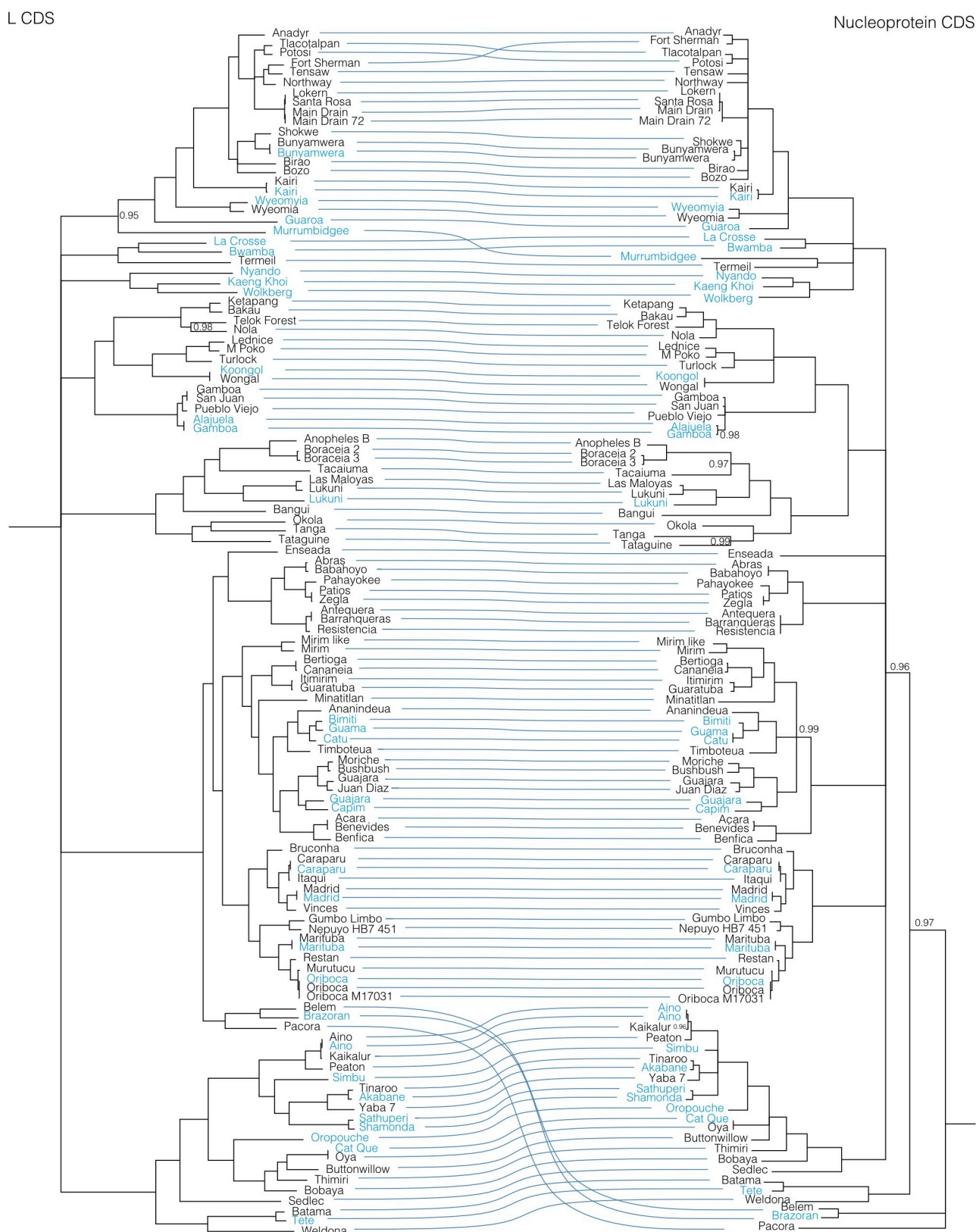

**Fig 7. Co-phylogenies of *Orthobunyavirus* L and S segment trees.** Branches with support values < 0.95 were converted to polytomies. All other branch support values were 1 unless indicated. Trees were rooted with outgroups as in Figs 2–4.

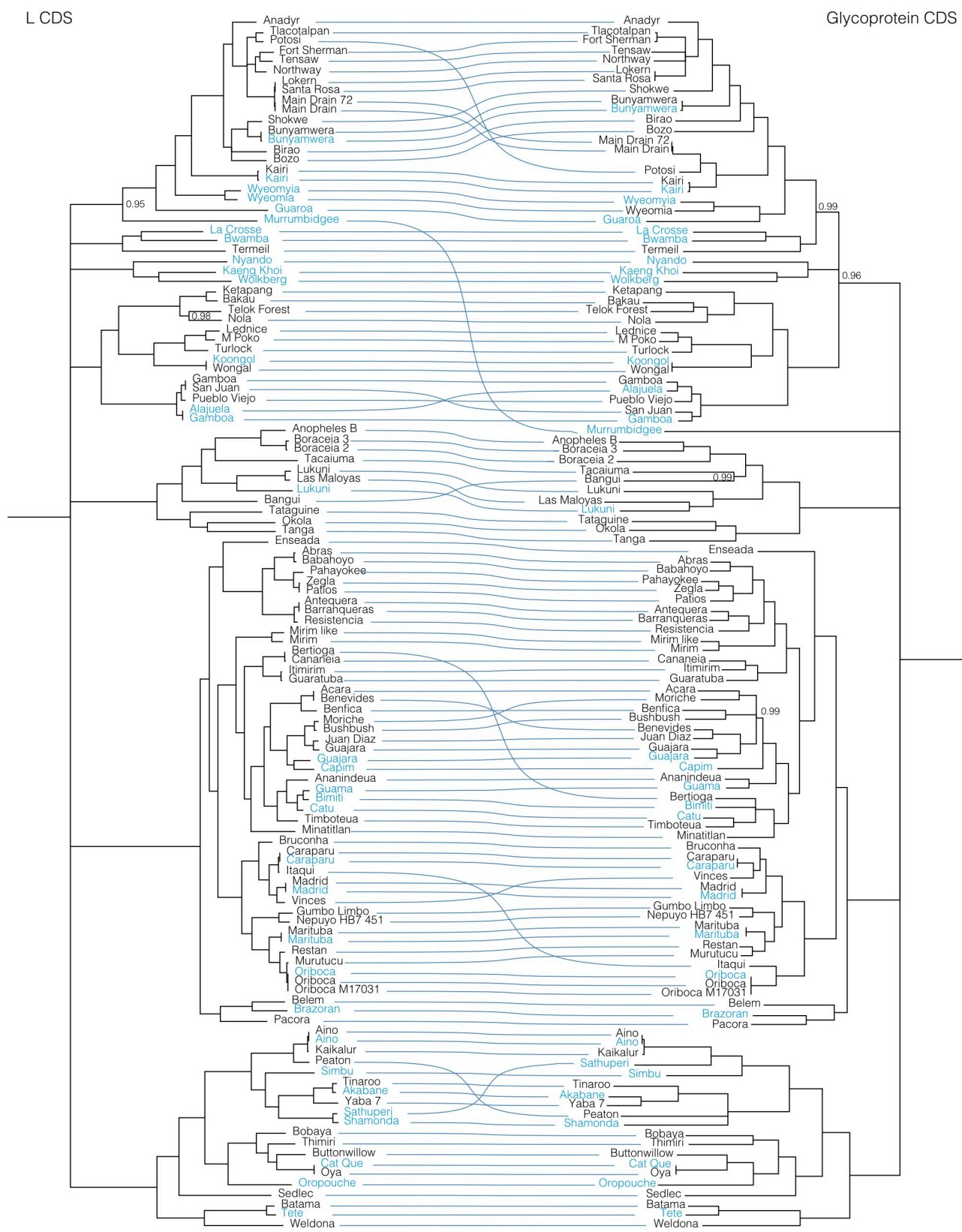

**Fig 8. Co-phylogenies of *Orthobunyavirus* L and M segment trees.** Branches with support values < 0.95 were converted to polytomies. All other branch support values were 1 unless indicated. Trees were rooted with outgroups as in Figs 2–4.

Nucleoprotein CDS

Glycoprotein CDS

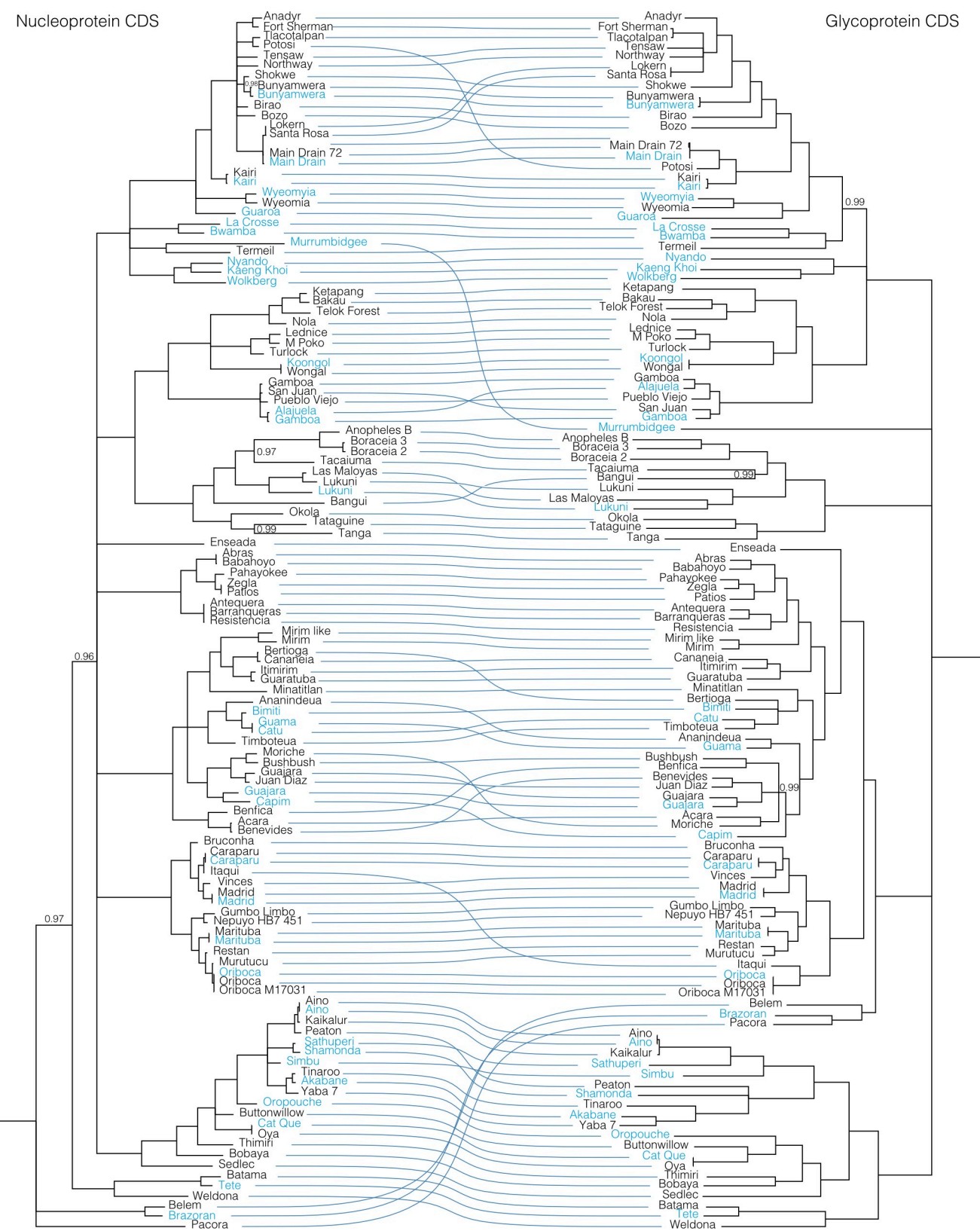

**Fig 9. Co-phylogenies of *Orthobunyavirus* S and M segment trees.** Branches with support values < 0.95 were converted to polytomies. All other branch support values were 1 unless indicated. Trees were rooted with outgroups as in Figs 2–4.

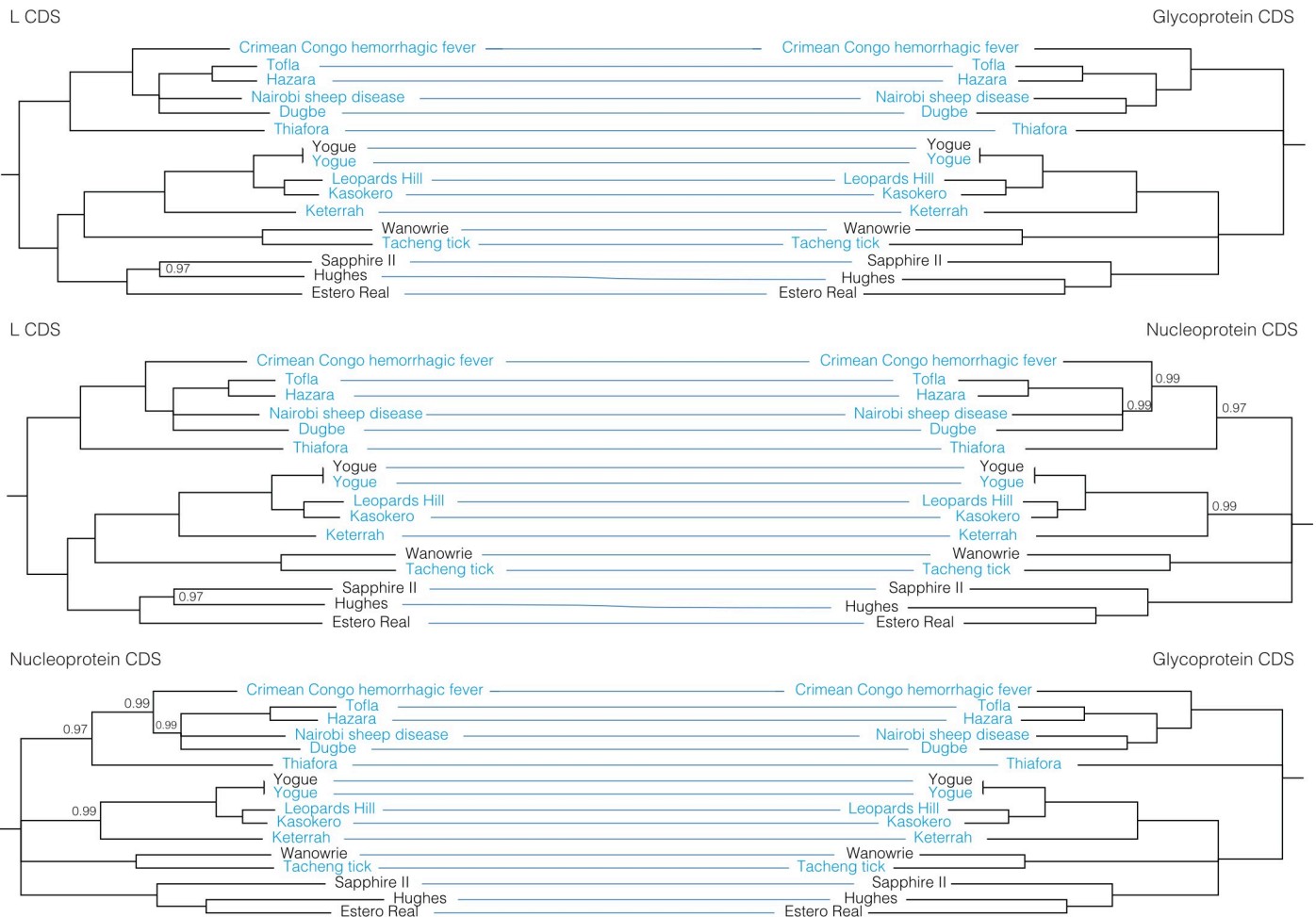

**Fig 10. Co-phylogenies of *Nairoviridae* L, M, and S segment trees.** Branches with support values < 0.95 were converted to polytomies. All other branch support values were 1 unless indicated. Trees were rooted with outgroups as in Fig 5.

serogroup most closely related to Mirim virus, with which it shared 77–81% pairwise nucleotide identity. Abras virus was present at a much higher abundance: 33,462 reads mapped to the three Abras virus segments, whereas only 2,091 reads mapped to the Mirim-like segments. Using RT-PCR with segment-specific primers (**S1 Table**), and RNA re-extracted from another vial of the isolate, we successfully amplified all six segments to confirm the coinfection. Additionally, we attempted to plaque-purify both viruses from the original isolate by Vero cell plaque assay. However, among the 90 viral plaques screened, none purely represented the co-infecting Guama serogroup virus, and passage of the co-infecting virus on Vero cells in an attempt to isolate it repeatedly failed. Additional preparations of Abras virus in the ARC that had been passaged in suckling mouse brain did not contain this co-infecting virus, leading us to suspect that the contaminant was introduced during original preparation of this particular virus stock that was sequenced during this study. Notably, Aguilar et al. (2018) did not report a co-infecting virus after also sequencing Abras isolate 75V1183 [137].

The Guaratuba virus dataset also produced 6 coding complete or partial bunyavirus segment sequences. Coding-complete sequences were assembled for all three segments of Guaratuba virus, as inferred based on their phylogenetic placement with other Guama serogroup viruses. An additional L, M, and S segment were assembled, but only the S segment was

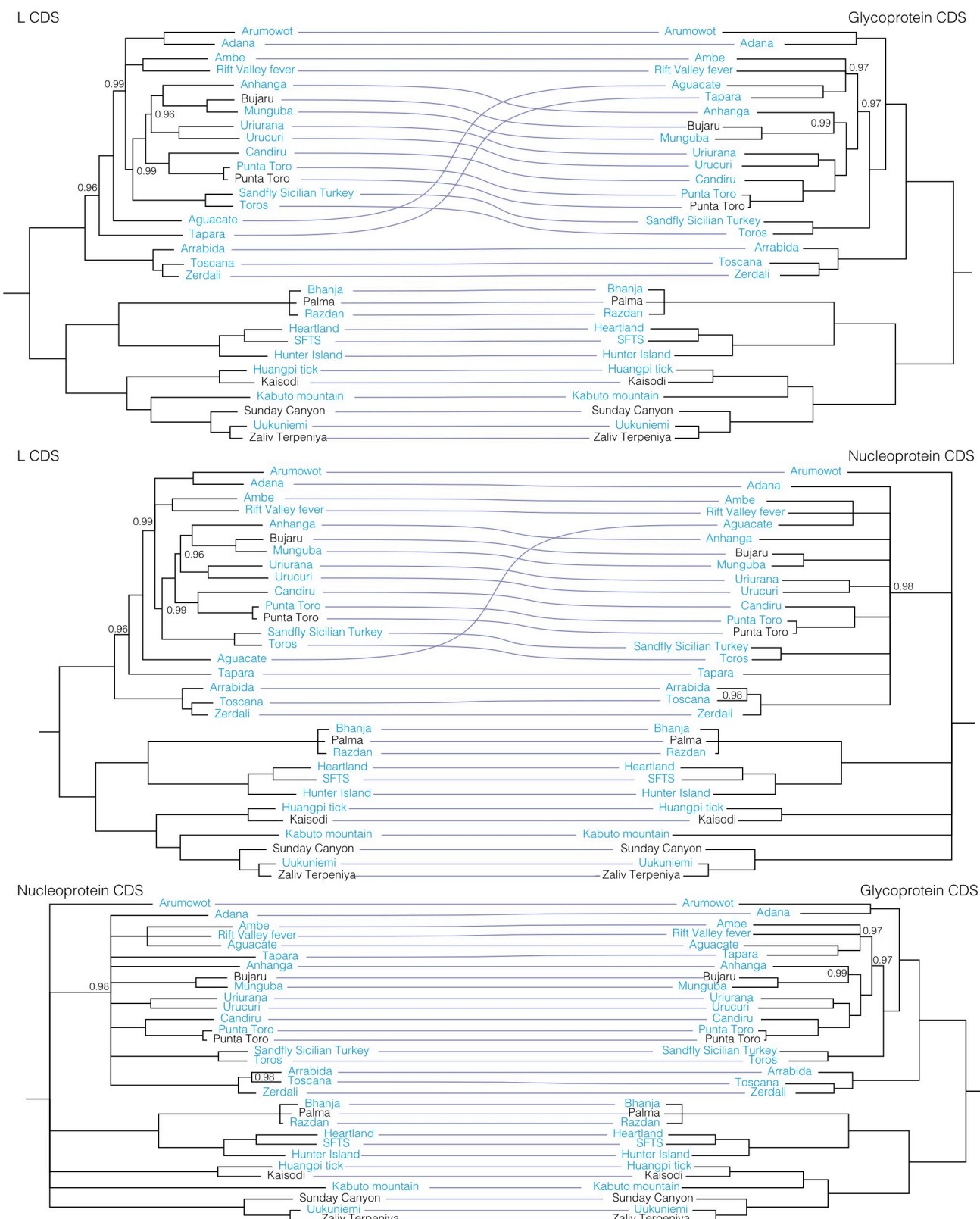

**Fig 11. Co-phylogenies of *Phenuiviridae* L, M, and S segment trees.** Branches with support values < 0.95 were converted to polytomies. All other branch support values were 1 unless indicated. Trees were rooted with outgroups as in Fig 6.

coding-complete. The additional partial L, M, and S sequences were ≈98% identical at the nucleotide level to strain 76V-25880 of Enseada virus [50,138]. The Guaratuba virus sequences had ≈7x more mapping reads (3,164) than the coinfecting Enseada-like virus (457). The Enseada virus-like sequences from the Guaratuba datasets were only 85–93% identical to the Enseada virus isolate strain 78V-213 genome that we sequenced for this project, indicating that these sequences were not attributable to cross-contamination during sample processing and library preparation. We used RT-qPCR to confirm the presence of all six segments in the original RNA from which the sequencing library was made. An independent RNA extract from Vero cells infected with Guaratuba virus from a second vial from the same lot was also positive for all 6 segments by RT-qPCR.

The Hughes virus dataset produced 4 bunyavirus contigs corresponding to a complete L, M, and S segment, and a partial M segment of about 1.2 kb. This partial M segment was nearly identical to a previously published Hughes virus M segment sequence (99.4% identical to strain DT-1, accession KP792739.1) The full length M sequence was only 81% identical to Hughes virus strain DT-1 and 72% identical to Hughes virus strain G2126 (accession KU925471). The partial M sequence had a coverage level of 2.4% relative to the full-length M sequence.

For Main Drain virus strain 72V2567, we assembled coding complete sequences for the L, M, and S segments with 452,386 mapping reads. In addition, 90 reads mapped to Cache Valley virus strain 6V633. We were able to corroborate the detection of the Cache Valley virus-like sequences using RT-qPCR with discriminating primers (**S1 Table**). The L, M, and S partial sequences were 98.5%, 95.9%, and 99% identical to Cache Valley virus respectively (MH166879.1, MH166878.1, MH166877.1).

## Discussion

Bunyaviruses comprise one of the largest viral orders, and some bunyaviruses pose substantial threats to humans, animals, and plants (3). We generated complete genome sequences for 99 bunyaviruses belonging to the families *Nairoviridae*, *Peribunyaviridae*, and *Phenuiviridae*, and analyzed them phylogenetically. Our results were consistent with recently published phylogenies on the Patois serogroup viruses [137] and a diversity of orthobunyaviruses in the Anopheles A, Capim, Guamá, koongol, Mapputta, Tete, and Turlock serogroups [129], but difficult to compare entirely given the inclusion of different viruses in each published analysis. Given the large number of viruses sequenced in this study, we elected to limit additional sequences in our phylogenies to new sequences plus those in NCBI RefSeq.

The phylogenies recapitulated assigned serogroups relatively well, however there were several points of inconsistency between the two. It is not that surprising that the determined phylogenetic relationships did not completely recapitulate established serogroups. Serology is based on similarities/dissimilarities in the surface proteins of the virion particle (encoded exclusively by M segments), whereas phylogenetic analysis can assess the relationships between any genes of any viral genome segment. Although one might expect the M segment to recapitulate the classical serogroup distinctions, this was not the case (**Figs 3, 5B, and 6B**). The L segment is the most conserved segment among all three families and most closely recapitulates the classical serogroup distinctions (**Figs 2, 5A, and 6A**). However, we found that several serogroups were not monophyletic in L segment phylogenies. For example, the Guama serogroup consisted of polyphyletic clades on all segments (**Figs 2–4**). Capim and Patois serogroups were also polyphyletic in the S segment tree (**Figs 4, 5C, and 6C**). Additionally, we were unable to resolve several ancestral branches which resulted in several polytomies, particularly in orthobunyavirus trees (**Fig 4**).

Many human pathogenic viruses in the genus *Orthobunyavirus* are found within the California, Bunyamwera, Simbu, and Group C serogroups. To these serogroups we have collectively added 33 coding-complete genome sequences. Although the pathogenic potential remains to be determined for many of the viruses sequenced in this study, viruses falling within serogroups for which there were previously no genomic data available will facilitate the identification and prioritization of novel potentially pathogenic strains for further study.

## Reassortment

These new sequences also contribute to a richer database with which to resolve the origins of reassortant viruses. Briese et al. [17] proposed that many, if not all, currently recognized bunyaviruses are the product of recent or ancient reassortants between known and/or possibly extinct viruses. High-level analysis for evidence of reassortment supported this hypothesis, and the conclusion that reassortment is a major factor driving bunyavirus genetic variability particularly in the Bunyamwera, Capim, and Group C serogroups (**Figs 7–9**). Reassortment among bunyaviruses has the potential to result in the emergence of novel human pathogens, as has been the case with Iquitos virus [9], Itaya virus [13], and Ngari virus [139].

L and S segment trees were largely concordant, and reassortment was, in almost all cases, evident as pairs or groups of viruses that had similar L and S segments but different M segments. Previous studies have reported that reassortant bunyaviruses tend to combine the S and L segments from one parent with the M segment from the other parent, making this a seemingly common phenomenon [19,139–141], although this is not always the case [17]. Linkage disequilibrium analysis of LACV isolates from field-collected mosquitoes showed evidence of frequent reassortment among strains in the field, and among all three genome segments [130]. Among the viruses we analyzed, there were several examples of pairs of viruses that had nearly identical S segments but relatively different L and M segments, suggesting recent reassortment events that may have involved the joining of the S segment of one parental virus with the L and M segment of the second parent. The reassortment that has happened frequently throughout bunyavirus evolution has introduced phylogenetic scrambling that makes it difficult to tease apart exact relationships. Sequencing and analysis of larger numbers of bunyaviruses will continue to shed light on the reassortment that has driven and continues to shape bunyavirus evolution.

## Co-infection of viral stocks

We also uncovered evidence of co-infections present in four virus stocks. It is possible that multiple viruses could be co-isolates from pools of mosquitoes containing multiple viruses. Overlapping ranges of the co-infecting viruses could provide support for this hypothesis. For instance, we detected Cache Valley virus reads in the Main Drain strain 72V2567 isolate. This Main Drain virus stock was derived from *Aedes vexans* mosquitoes collected in New Mexico, USA, in 1972, and Cache Valley virus is broadly distributed in Northern and Central America [142]. Similarly, there was evidence of an Enseada virus co-infection in the Guaratuba virus isolate that we sequenced. Guaratuba and Enseada virus have both been isolated from pools of *Culex* mosquitoes collected in 1976 in Brazil [50,138].

Ultimately, however, the true origins of these apparent co-infections are not knowable from our data alone. It is possible that cross-contamination may have occurred during isolation or passage. The failure to identify the co-infecting Mirim-like virus in an independent stock of Abras virus supports this alternative in that case. We can exclude cross-contamination during RNA extraction, library preparation, and sequencing because the co-infecting viruses were not identical to any of the other viruses we sequenced, and we confirmed the presence of co-

infecting RNAs in separate vials of a stock for the Abras, Hughes, and Guaratuba virus isolates. Testing of the original samples and resequencing earlier passages of a virus could resolve ambiguity about the origin of co-infections, but it is likely that the source material for most of these isolates is no longer available. Continuing surveillance and direct sequencing of virus genomes in field samples without virus passage in cell culture will provide additional insight into the extent to which individual vectors or vector populations harbor bunyavirus co-infections, and the extent to which this impacts reassortment potential.

## Taxonomic implications

The recent expansion in sequenced bunyaviral genomes–and of RNA virus genomes in general–has led to a restructuring of bunyavirus taxonomy [14,15,143–146]. Some of the viruses sequenced in this study will undoubtedly lead to the establishment of new genera and species, particularly within the *Orthobunyavirus* genus. Our data highlight some unresolved issues with bunyavirus taxonomy that are attributable to pervasive reassortment. For instance, the question of how should pairs of viruses that are nearly identical in 2 of the 3 genome segments but highly divergent in the 3rd, like Bertioga and Cananéia viruses, or Santa Rosa and Main Drain viruses, be classified?

## Conclusions

This study contributed 35 totally new bunyavirus genome sequences to the public domain. These sequences further enrich the reference data available for the identification of emerging bunyaviruses, facilitate the resolution of phylogenetic relationships among known and newly-described viruses, and provide additional context towards the identification of reassortant strains. In addition, the generation of such a large dataset permitted expanded analyses of co-infection and reassortment. Each of these analyses provided foundational data which will support future investigations on the genetic diversity, reassortment, and virus-vector-host interactions in this important group of emerging human pathogens.

## Supporting information

**S1 Fig. Coverage depth across all genome segments for all sequenced viruses.** Coverage represents mean coverage for non-overlapping 10 nucleotide windows.
(PDF)

**S2 Fig. Tree containing all available *Orthobunyavirus* sequences.** All complete L protein sequences annotated under the *Orthobunyavirus* genus in the NCBI Taxonomy database were downloaded and used to infer a maximum likelihood phylogeny. Sequences that we generated in the course of this study are indicated in red. Tree is midpoint rooted. Scale bar indicates substitutions per site.
(PDF)

**S3 Fig. Tree containing all available *Phlebovirus* sequences.** All complete L protein sequences annotated under the *Phlebovirus* genus in the NCBI Taxonomy database were downloaded and used to infer a maximum likelihood phylogeny. Sequences that we generated in the course of this study are indicated in red. Tree is midpoint rooted. Scale bar indicates substitutions per site.
(PDF)

**S4 Fig. Tree containing all available *Nairoviridae* sequences.** All complete L protein sequences annotated under the *Nairoviridae* family in the NCBI Taxonomy database were

downloaded and used to infer a maximum likelihood phylogeny. Sequences that we generated in the course of this study are indicated in red. Tree is midpoint rooted. Scale bar indicates substitutions per site.
(PDF)

**S5 Fig. L segment phylogeny for viruses in *Peribunyaviridae*: *Pacuvirus*.** The virus sequences we generated in this study are shown in black.
(PDF)

**S1 Table. Primers used in this study.**
(XLSX)

**S2 Table. Pairwise nucleotide identities from a global alignment of the L, GPC, or N coding sequences.** Virus names with _reference in them derive from RefSeq sequences.
(XLSX)

## Acknowledgments

We thank Dr. Amy Lambert for project support and technical guidance.

## Author Contributions

**Conceptualization:** Rebekah C. Kading, Mark D. Stenglein.

**Data curation:** Marylee L. Kapuscinski, Justin S. Lee, Mark D. Stenglein.

**Formal analysis:** Marylee L. Kapuscinski, Nicholas A. Bergren, Justin S. Lee, Daniel A. Hartman, Rebekah C. Kading, Mark D. Stenglein.

**Funding acquisition:** Rebekah C. Kading, Mark D. Stenglein.

**Project administration:** Rebekah C. Kading, Mark D. Stenglein.

**Resources:** Brandy J. Russell, Holly R. Hughes.

**Software:** David C. King.

**Supervision:** Rebekah C. Kading, Mark D. Stenglein.

**Validation:** Marylee L. Kapuscinski, Nicholas A. Bergren, Brandy J. Russell, Justin S. Lee, Erin M. Borland, Holly R. Hughes, Kristen L. Burkhalter, Rebekah C. Kading, Mark D. Stenglein.

**Writing – original draft:** Marylee L. Kapuscinski, Nicholas A. Bergren, Rebekah C. Kading, Mark D. Stenglein.

**Writing – review & editing:** Marylee L. Kapuscinski, Nicholas A. Bergren, Brandy J. Russell, Justin S. Lee, Erin M. Borland, Holly R. Hughes, Kristen L. Burkhalter, Rebekah C. Kading, Mark D. Stenglein.

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
