## [Decision Letter · Decision Letter 0]

13 Oct 2020

Dear Dr. Stenglein,

Thank you very much for submitting your manuscript "Genomic Characterization of 99 viruses from the Bunyaviral Families Nairoviridae, Peribunyaviridae, and Phenuiviridae" for consideration at PLOS Pathogens. As with all papers reviewed by the journal, your manuscript was reviewed by members of the editorial board and by several independent reviewers. In light of the reviews (below this email), we would like to invite the resubmission of a significantly-revised version (text changes) that takes into account the reviewers' comments.

We cannot make any decision about publication until we have seen the revised manuscript and your response to the reviewers' comments. Your revised manuscript is also likely to be sent to reviewers for further evaluation.

Sincerely,

Jens H. Kuhn

Associate Editor

PLOS Pathogens

David Wang

Section Editor

PLOS Pathogens

Kasturi Haldar

Editor-in-Chief

PLOS Pathogens

orcid.org/0000-0001-5065-158X

Michael Malim

Editor-in-Chief

PLOS Pathogens

orcid.org/0000-0002-7699-2064

Reviewer's Responses to Questions

**Part I - Summary**

Reviewer #1: This is an incredibly impressive manuscript where the authors sequenced and phylogenetically analyzed the complete ORFs of 99 bunyaviruses in the CDC’s archived collections. The viruses were collected over 66 years in multiple countries. I was particularly impressed with Figures 7 to 11 which showed all of ressortment events. I have never before seen ressortment data presented in this way before. All of my concerns are minor.

It was not immediately clear to me as to how many novel viruses were identified because they had previously been misidentified. Considering mentioning this in the abstract or towards the start of the discussion.

It could be worthwhile to mention the ICTV’s criteria for species demarcation (optional)

Page 35 explains that reassortment events usually arouse from M segment swaps. Others have reported which segment swaps occur most often. This prior work should be mentioned in the discussion

Line 225 – I suggest deleting (N=2) and (N=6) from the text and instead say “1940 t0 2005”.

Page 1 –Table 1 – so references are italicized, others are not

Whenever “pairwise identity” is mentioned, it would be preferable if it were “pairwise nucleotide identity” to avoid confusion with “amino acid identity”

Reviewer #2: Kapuscinski and colleagues have sequenced several new viral genomes and supplemented sequence fragments to previously described arboviral species from a wide range of vertebrates and arthropods. The authors used a combination of Illumina- and Sanger-based sequencing approaches on a large repository of novel and ARC bunyaviruses. In some cases, plaque purifications were necessary to separate coinfections in vitro. Phylogenetic analyses using a Bayesian approach helped frame the sequencing contributions in an evolutionary and serological context. In particular, the co-phylogenies are well presented and bring attention to bunyavirus reassortment events.

Reviewer #3: Kapuscinski and colleagues present a comprehensive study in which they determined coding complete sequences of 99 cell cultured Bunyavirus isolates. 35 of these coding complete sequences may represent new Bunyavirales species. For sure an interesting and important study, but mainly descriptive with a lot of data to comprehend. Data which is not always presented in a clear way, Apart from that, there is actually almost no comment to give on this manuscript. Interesting findings, methods are clearly defined and appropriate. Results and Discussion are substantial and sometimes difficult to follow.

**Part II – Major Issues: Key Experiments Required for Acceptance**

Reviewer #1: All of my comments are noted above

Reviewer #2: Although there is a notable amount of novel genomic sequences reported in the manuscript, the authors may be unintentionally "overselling" their results in two main ways:

First, the number of “real” new viruses provided in this manuscript is 35 (counting “1” in Tables 1-3). This number is not mentioned anywhere in the text. If we count the 14 number “3”s in the same tables, it still is not even close to the consistently reported “99 viruses” that the authors present. The other numbers and the specific contributions of the authors should be elaborated in the text and tables. For example, in the case of “2”, does this equate to new metadata on the sample, or did the authors sequence a new isolate of an already catalogued virus while also adding some additional information? Looking at Bunyamwera virus from 2006, the complete genome sequence was deposited two years ago by different authors (MH484288-90), so I am not sure what was contributed from a genomic standpoint in this study. But then for Enseada virus, the strain name differs than the existing entries in GenBank and this seems like a new addition to Enseada virus genomic diversity.

For “4” why are these included in the “99” if they have already been fully characterized elsewhere? The scientific merit and relevance of including resequenced strains with already existing full genomes is not made clear here. This could be important is if the authors are using a primary sample or earlier cell passage than the current sequenced isolate. This would provide a more “authentic” genomic sequence, which could have evolutionary and possibly medical countermeasure implications. The authors should also consider directly linking new and existing viral isolates to the ARC.

Second, the estimated phylogenetic trees were complemented with publicly available sequences only from RefSeq, which is also “oversells" the true extent of the author’s contributions. This approach makes the author’s “new” black entries look overwhelmingly high compared with the blue entries (the ones previously sequenced). The authors should change the conclusions to better reflect the novel results of their presented work. The trees should include all available complete ORF sequences in GenBank/SRA to highlight existing bunyavirus diversity. This will provide a greater framework for the author’s contributions.

Furthermore, there are numerous instances where credit was not given to the appropriate publication, particularly if the virus was completely or partially sequenced previously. While acknowledging the original discovery of a particular virus or serogroup is important, sequencing efforts have occurred globally over the last decade and should be equally acknowledged for their respective contributions.

Overall, I do think there are some important characterizations and new information that advance the field. However, the author’s claims need to be reassessed and clarified in several places.

Reviewer #3: I do not have any major issues.

**Part III – Minor Issues: Editorial and Data Presentation Modifications**

Reviewer #1: All of my comments are noted above

Reviewer #2: Other minor comments:

Line 100: “Viruses represented a subset of bunyaviruses catalogued within the ARC (22) (Tables 1-3).”

As mentioned above, it would be helpful to provide links to the ARC in the main tables for each virus listed. In the cases where new information was added, specific contributions should be elaborated.

Line 109: “The KAPA RNA HyperPrep Kit was used to prepare sequencing libraries from total RNA according to the manufacturer’s protocol using half-scale reactions (Kapa Biosystems).”

What type of adapters were used? What measures were taken to limit adapter bleedthrough and ensure that reassortments were not as a result of laboratory contamination? In the Discussion, the author’s say that contamination has been ruled out, but they do not provide specific details and safeguards to ensure this.

Line 124: “Viral contigs were further assembled using Geneious v11.0.2 as necessary (32).”

What settings were used in Geneious? Did you have a minimum depth cutoff? How were consensus sequences determined in low coverage areas? You mention the 4x cutoff at the ends, is that used throughout? If so, how were disagreements handled e.g. four total reads mapping two Adenines and two Guanines at the same position?

Line 132: “…and alignments were independently validated by two people.”

I do not understand the significance of a two person validation.

Line 172: “Poorly aligned, divergent positions characterized by excessive gaps in the alignment were removed using the Gblocks server under less strict conditions (37).”

First, I do not see any link or access point to alignments, which are crucial to the later phylogenetic inferences. The authors stated that all data should be open and should follow this. Second, I don’t think removing diverging positions or gaps is correct. Each site plays a role in reconstructing phylogenetic relationships and they should not be removed, especially in comparisons among distinct viral species.

Line 216: “Single virus stocks were cultured from plaques that were sequence-confirmed to have only one of the infecting genotypes present.”

How were they sequence confirmed?

Line 235: “Some viruses were passaged through grivet (Chlorocebus aethiops) Vero or hamster (Mesocricetus auratus) BHK-21 cell cultures, others were passaged in suckling laboratory mice.”

Referring to an earlier comment, it would be nice to have passage histories available of the newly discovered virus stocks generated as part of this study.

Line 231/Figure 1E: “203 of the genome segment sequences we generated shared less than 97% pairwise nucleotide identity with existing sequences (Figure 1E).”

The authors discuss a lot of taxonomical implications in the manuscript. They may consider expanding this section by including a DEmARC analysis or other taxonomically related tool to differentiate new species and genera.

Lines 280-287. I think this paragraph should be moved and merged into the Methods section. There is also some information here that isn’t explicitly stated in the actual Methods.

Line 285: “…which produced 748-fold mean coverage depth across virus genomes.”

The authors should include the coverage depth or possibly coverage plots for each virus somewhere in the manuscript or supplementary materials.

Line 326: “This observation supports the hypothesis that this group of viruses resulted from an ancestral reassortment event involving the replacement of an S segment, as has been proposed for Brazoran virus (117).”

This is more of a discussion point than a result. I would move it to the Discussion or remove it.

Line 422: “In the Bakau serorgroup…”

Serogroup?

Line 437: “The Tinaroo and Akabane virus L and S segments were relatively closely related, but the M segments shared only 65% pairwise identity.”

I would add a percentage or some quantification to “closely related” to match the last part of the sentence.

Line 461: “There was little evidence for reassortment among the orthonairoviruses in our analysis.”

Why did the authors opt-out of using the RDP4 package to test for reassortment? I am not disagreeing with the author’s approach, but RDP software has been used previously to confirm reassortments in bunyaviruses.

Line 543: “The partial M sequence had a normalized coverage level of 2.4% relative to the full-length M sequence.”

Please clarify the meaning of this statement.

Reviewer #3: It is in the nature of a report of this type to focus on perceived shortcomings and incompleteness. My comments should be seen in the context of improving an already good manuscript.

* lines 100-117: I was surprized no additional depletion steps were included apart from DNA depletion after RNA extraction for suckling mouse brain preparations. The KAPA RNA HyperPrep kit was used for both sample types without rRNA or mRNA depletion steps, is that correct?

* I realise that Tables 1 to 3 are already extensive, but it would be good adding the corresponding GenBank accession numbers here as well. Also mark if segments are coding complete or full-lengths. It is actually confusing that most GenBank entries are marked with "complete sequence" but are apparently coding complete.

If available, a strain passage number before sequencing would be of interest as well... and overall coverage of segments.

* GenBank entries also miss sampling date, species of origin, sampling location, ...

* For several of the isolates, 1 or more segments were verified with Sanger sequencing. What was the reason? Low coverage or incomplete assembly?

Did the authors attempted to determine the 5`and 3`panhandle structures and down/upstream regions? It is actually a shame that did was not done.

* line 169-170: This makes no sense. Amino acid sequences are aligned with muscle, after that 'manually aligned afterwards using Seaview'...? Should this read 'manually corrected'? If so, why a manual correction and what was corrected? otherwise, why align 2 times?

Why are alignments converted back to nucleotides after aa alignments. How was this done and was the original nt sequence used or a generic codons? Why not use a better multiple alignment algorithm that can work better with divergent datasets and use the amino acid alignment as a guide for nucleotide alignment.

* For the Bayesian analysis performed, it is unnecessary to include an outgroup in this type of analysis. This can also introduce unnecessary disturbance and excessive gaps in the alignment.

* line 187-216. Another strategy would be looking at variability in 5' and 3' panhandle structures.

* Define scale bar in phylogenetic tree figure legends.

* Figure 5: which hantavirus was used as outgroup?

* The manuscript describes a lot of data. It is not always easy and clear to follow the results and discussion sections. Maybe consider the use subheadings in both sections.

PLOS authors have the option to publish the peer review history of their article (what does this mean?). If published, this will include your full peer review and any attached files.

Reviewer #1: No

Reviewer #2: No

Reviewer #3: No
---

## [Editor Report · Decision Letter 1]

13 Jan 2021

Dear Dr. Stenglein,

We are pleased to inform you that your manuscript 'Genomic Characterization of 99 viruses  from the Bunyaviral Families Nairoviridae, Peribunyaviridae, and Phenuiviridae, including 35 previously unsequenced viruses.' has been provisionally accepted for publication in PLOS Pathogens.

Best regards,

Jens H. Kuhn

Associate Editor

PLOS Pathogens

David Wang

Section Editor

PLOS Pathogens

Kasturi Haldar

Editor-in-Chief

PLOS Pathogens

orcid.org/0000-0001-5065-158X

Michael Malim

Editor-in-Chief

PLOS Pathogens

orcid.org/0000-0002-7699-2064
---

## [Editor Report · Acceptance letter]

23 Feb 2021

Dear Dr. Stenglein,

We are delighted to inform you that your manuscript, "Genomic characterization of 99 viruses  from the bunyaviral families *Nairoviridae *, *Peribunyaviridae *, and *Phenuiviridae*, including 35 previously unsequenced viruses," has been formally accepted for publication in PLOS Pathogens.

Best regards,

Kasturi Haldar

Editor-in-Chief

PLOS Pathogens

orcid.org/0000-0001-5065-158X

Michael Malim

Editor-in-Chief

PLOS Pathogens

orcid.org/0000-0002-7699-2064